# COMAL: A CONVERGENT META-ALGORITHM FOR ALIGNING LLMS WITH GENERAL PREFERENCES

## ABSTRACT

Many alignment methods, including reinforcement learning from human feedback (RLHF), rely on the Bradley-Terry reward assumption, which is insufficient to capture the full range of general human preferences. To achieve robust alignment with general preferences, we model the alignment problem as a two-player zero-sum game, where the Nash equilibrium policy guarantees a 50% win rate against any competing policy. However, previous algorithms for finding the Nash policy either diverge or converge to a Nash policy in a modified game, even in a simple synthetic setting, thereby failing to maintain the 50% win rate guarantee against all other policies. We propose a meta-algorithm, **Co**nvergent **M**eta **Al**ignment Algorithm (COMAL), for language model alignment with general preferences, inspired by convergent algorithms in game theory. Theoretically, we prove that our meta-algorithm converges to an exact Nash policy. Additionally, our meta-algorithm is simple and can be integrated with many existing methods designed for RLHF and preference optimization with minimal changes. Experimental results demonstrate the effectiveness of the proposed framework when combined with existing preference policy optimization methods.

## 1 INTRODUCTION

Large Language Models (LLMs) (Brown et al., 2020; OpenAI, 2023; Dubey et al., 2024) have fundamentally transformed the fields of natural language processing and artificial intelligence. They excel in tasks ranging from text generation and translation to complex question answering and interactive dialogue systems. As these models become more integrated into daily life, a key challenge is ensuring they achieve high levels of alignment with human values and preferences.

One of the most widely adopted approaches to addressing this challenge is Reinforcement Learning from Human Feedback (RLHF) (Christiano et al., 2017; Ouyang et al., 2022). This framework consists of two steps: first, learning a reward model from a dataset containing human preferences, and second, optimizing the LLM using the proximal policy optimization (PPO) algorithm (Schulman et al., 2017). Recently, Rafailov et al. (2024) observed that the first step can be bypassed, proposing the direct preference optimization (DPO) algorithm, directly optimizing the LLM from the dataset.

However, the aforementioned approaches crucially rely on the assumption that human preferences can be expressed using the Bradley-Terry (BT) model (Bradley and Terry, 1952). Unfortunately, the BT model is too restrictive to capture the richness and complexity of human preferences. Specifically, the BT model can only induce *transitive* preferences—i.e., if more people favor A over B, and B over C, then more people must favor A over C. Such transitivity may not hold in the presence of diverse populations and is also incompatible with evidence from human decision-making (May, 1954; Tversky, 1969).

To overcome this limitation, recent research has begun to explore alignment under general preferences. Munos et al. (2024) formulate this alignment problem as a symmetric two-player zero-sum game, where both players' strategies are LLMs, and their payoffs are determined by the win rate against the opponent's LLM according to the preference model. The objective is to identify a Nash equilibrium policy that guarantees at least a 50% win rate against any other policy (Azar et al., 2024; Munos et al., 2024; Calandriello et al., 2024), a property we refer to as *robust alignment*. However, all the proposed algorithms either diverge or converge to the Nash policy of a modified game, thereby failing to maintain the 50% win rate guarantee against all other policies.

**Our Contribution.** We introduce a novel meta-algorithm, **Co**vergent **M**eta **Al**ignment **Al**gorithm (COMAL), inspired by the conceptual prox-method, a convergent algorithm for solving two-player zero-sum games (Nemirovski, 2004). Our first observation is that many existing algorithms, including PPO (Schulman et al., 2017), DPO (Rafailov et al., 2024), IPO (Azar et al., 2024), SPPO (Wu et al., 2024), INPO (Zhang et al., 2024), etc., can be interpreted as implementations of the Prox operator (Nemirovski, 2004). COMAL employs the Prox operator as its fundamental building block and provably *converges* to the Nash equilibrium policy in the *last iterate*, assuming the Prox operator can be computed exactly, thus achieving *robust alignment*. This approach allows us to leverage many existing algorithms in a black-box manner. While several algorithms, e.g., IPO, SPPO, etc., in the literature demonstrate average-iterate convergence to the Nash equilibrium policy, they all diverge in the last iterate. Unfortunately, iterate averaging can be cumbersome, particularly when deep-learning components are involved, as it may not be feasible to average the outputs of LLMs.[1] Compared to these algorithms, COMAL achieves the more desirable last-iterate convergence.

In addition to our theoretical analysis, we validate the effectiveness of COMAL through both synthetic and LLM-based experiments.

**Synthetic experiments.** We construct a $3 \times 3$ two-player zero-sum preference game, and compare COMAL with a wide range of algorithms proposed in the literature. The result clearly shows that COMAL is the only algorithm that converges to the Nash equilibrium of the game in the last iterate.

**LLM-based experiments.** Furthermore, we evaluate the performance of COMAL against existing preference optimization algorithms under a real-world setting, where a pre-trained LLM, Qwen2-1.5B (Yang et al., 2024), is fine-tuned using different algorithms on the UltraFeedback (Cui et al., 2023) dataset, which is commonly used for alignment fine-tuning of LLMs. Our experimental results demonstrate the advantages of COMAL: it achieves at least 55% win rate compared against baseline algorithms including DPO (Rafailov et al., 2024) and iterative algorithms such as iterative IPO (Azar et al., 2024) and INPO (Zhang et al., 2024).

## 2 BACKGROUND

We use $\Delta(\mathcal{Z})$ to denote a distribution over a set $\mathcal{Z}$. We denote $x \in \mathcal{X}$ as an instruction where $\mathcal{X}$ is the instruction set. We assume a fixed distribution $\rho \in \Delta(\mathcal{X})$ over the instruction set. We denote $\mathcal{Y}$ as the response set and $y \in \mathcal{Y}$ as one response. Given any instruction $x \in \mathcal{X}$, an LLM policy $\pi$ specifies the output distribution $\pi(\cdot \mid x) \in \Delta(\mathcal{Y})$. For distributions $p, q \in \Delta(\mathcal{Z})$, the Kullback-Leibler (KL) divergence is defined as $\mathrm{KL}(p||q) := \sum_{z \in \mathcal{Z}} p(z) \log \frac{p(z)}{q(z)}$. The sigmoid function is $\sigma(x) := \frac{e^x}{e^x + 1}$. We use $\mathrm{supp}(p)$ to denote the support of a distribution $p$.

**Preference Models** In this paper, we focus on general preference models.

**Definition 1** (General Preference Model). *A general preference model* $\mathbb{P} : \mathcal{X} \times \mathcal{Y} \times \mathcal{Y} \to [0, 1]$ *satisfies* $\mathbb{P}(y_1 \succ y_2 \mid x) = 1 - \mathbb{P}(y_2 \succ y_1 \mid x)$. *When we query* $\mathbb{P}$ *with* $(x, y_1, y_2)$, *it outputs* 1 *with probability* $\mathbb{P}(y_1 \succ y_2 \mid x)$ *meaning* $y_1$ *is preferred over* $y_2$, *and it outputs* 0 *otherwise.*

We define $\mathbb{P}(\pi_1 \succ \pi_2) := \mathbb{E}_{x \sim \rho}[\mathbb{E}_{y_1 \sim \pi_1, y_2 \sim \pi_2}[\mathbb{P}(y_1 \succ y_2 \mid x)]]$ as the *win rate* of $\pi_1$ over $\pi_2$ under preference model $\mathbb{P}$. A special case of the general preference model is the Bradley-Terry (BT) model, which assumes a reward function parameterizes the preference.

**Definition 2** (Bradley-Terry Model). *A preference model* $\mathbb{P}$ *satisfies the* Bradley-Terry (BT) *assumption if there exists a reward function* $r^* : \mathcal{X} \times \mathcal{Y} \to \mathbb{R}$ *such that*

$$\mathbb{P}(y_1 \succ y_2 \mid x) = \frac{\exp\left(r^*(x, y_1)\right)}{\exp\left(r^*(x, y_1)\right) + \exp\left(r^*(x, y_2)\right)} = \sigma(r^*(x, y_1) - r^*(x, y_2)).$$

### 2.1 ALIGNMENT UNDER THE BRADLEY-TERRY MODEL ASSUMPTION

**RLHF** The canonical formulation of Reinforcement Learning from Human Feedback (RLHF) is to first learn a reward function $r$ under the BT model and then find the optimal KL regularized policy

---

[1]Storing all LLMs produced during training could solve this, but it is highly space-inefficient and, to our knowledge, has not been implemented.

$\pi^*$ with respect to the learned reward function $r$:

$$\pi^* := \arg\max_{\pi} \mathbb{E}_{x\sim\rho, y\sim\pi(\cdot|x)}\big[r(x,y) - \eta^{-1}\,\mathrm{KL}(\pi(\cdot\mid x)||\pi_{\mathrm{ref}}(\cdot\mid x))\big], \tag{1}$$

where $\eta^{-1} > 0$ controls the regularization, and $\pi_{\mathrm{ref}}$ is the initial reference model, usually the policy $\pi_{\mathrm{sft}}$ obtained from pre-training and supervised fine-tuning.

**DPO** Rafailov et al. (2024) observe that the regularized optimization problem (1) has a closed-form solution : for any $x$ and $y$,

$$\pi^*(y\mid x) = \frac{\pi_{\mathrm{ref}}(y\mid x)\exp\left(\eta r(x,y)\right)}{Z_x}, \tag{2}$$

where $Z_x = \mathbb{E}_{y\sim\pi_{\mathrm{ref}}(\cdot|x)}[\exp(\frac{1}{\eta}r(y,x))]$ is the normalization constant known as the partition function. In (2), we see that $\pi^*$ implicitly parameterizes the reward function $r$. Rafailov et al. (2024) propose direct preference optimization (DPO) to learn the optimal policy using the maximum likelihood objective directly:

$$\ell_{\mathrm{DPO}}(\pi;\pi_{\mathrm{ref}}) = -\mathbb{E}_{(x,y_w,y_l)\sim\mathcal{D}}\left[\log\sigma\left(\eta^{-1}\log\frac{\pi(y_w\mid x)}{\pi_{\mathrm{ref}}(y_w\mid x)} - \eta^{-1}\log\frac{\pi(y_l\mid x)}{\pi_{\mathrm{ref}}(y_l\mid x)}\right)\right],$$

where $\mathcal{D}$ is a data set containing win-loss pair of responses $\{y_w, y_l\}$ given prompt $x$.

## 2.2 ALIGNMENT WITH GENERAL PREFERENCE MODELS

The BT model assumption is insufficient to capture the full range of general human preferences (Munos et al., 2024; Swamy et al., 2024). To achieve robust alignment with general preferences, we model the policy optimization problem as a two-player zero-sum game with the objective function as follows:[2]

$$J(\pi_1, \pi_2) := \mathbb{P}(\pi_1 \succ \pi_2) - \frac{1}{2} = \mathbb{E}_{x\sim\rho}[\mathbb{E}_{y_1\sim\pi_1, y_2\sim\pi_2}[\mathbb{P}(y_1 \succ y_2 \mid x)]] - \frac{1}{2}. \tag{3}$$

In this game, the max-player controls $\pi_1$ and tries to maximize $J(\pi_1, \pi_2)$ while the min-player controls $\pi_2$ and tries to minimize $J(\pi_1, \pi_2)$. We focus only on policies with $\Pi := \{\pi : \mathrm{supp}(\pi) \subseteq \mathrm{supp}(\pi_{\mathrm{sft}})\}$ in the support of the initial SFT policy. A Nash equilibrium policy $(\pi_1^\star, \pi_2^\star)$ satisfies

$$\pi_1^\star, \pi_2^\star \in \underset{\pi_1\in\Pi}{\mathrm{argmax}}\,\underset{\pi_2\in\Pi}{\mathrm{argmin}}\, J(\pi_1, \pi_2), \quad J(\pi_1, \pi_2^\star) \leq J(\pi_1^\star, \pi_2^\star) \leq J(\pi_1^\star, \pi_2), \forall\pi_1, \pi_2 \in \Pi.$$

Since $J(\pi_1, \pi_2)$ is symmetric, the game has a symmetric Nash equilibrium $(\pi^\star, \pi^\star)$. Moreover, the Nash equilibrium policy $\pi^\star$ guarantees that for any other policy $\pi$, its win rate is at least $\mathbb{P}(\pi^\star \succ \pi) \geq \mathbb{P}(\pi^\star \succ \pi^\star) = 50\%$. We call this property *robust alignment*. Our goal is to find a policy with robust alignment.

Existing online iterative preference optimization methods designed for or applicable to the original game including iterative IPO (Azar et al., 2024) and SPPO (Wu et al., 2024), are based on Multiplicative Weights Update, and thus *diverge* as we show in Section 4. There is also a line of works including Nash-MD (Munos et al., 2024; Ye et al., 2024), Online IPO (Calandriello et al., 2024), INPO (Zhang et al., 2024) aim to find the Nash equilibrium of a modified KL-regularized game:

$$J_\tau(\pi_1, \pi_2, \pi_{\mathrm{ref}}) := J(\pi_1, \pi_2) - \tau\mathbb{E}_{x\sim\rho}[\mathrm{KL}(\pi_1(\cdot\mid x)||\pi_{\mathrm{ref}}(\cdot\mid x))] + \tau\mathbb{E}_{x\sim\rho}[\mathrm{KL}(\pi_2(\cdot\mid x)||\pi_{\mathrm{ref}}(\cdot\mid x))].$$

The additional KL regularization terms in the objective are introduced for training stability. However, the Nash equilibrium of the modified game no longer achieves robust alignment, i.e., it has a win rate of at least 50% against any competing policy. We present comparison of these algorithms in Table 1.

Moreover, most existing theoretical convergence guarantees only hold for the average iterate, i.e., the uniform mixture of training iterates, which is not used in practice. We focus on designing algorithms with provable last-iterate convergence to Nash equilibrium, which aligns with practice and is more space-efficient (Munos et al., 2024).

As we show in the next section, our meta-algorithm COMAL can also be implemented with black-box access to algorithms that solve the regularized game $J_\tau(\pi_1, \pi_2, \pi_{\mathrm{ref}})$.

---

[2]We introduce the constant $\frac{1}{2}$ only to ensure the game is zero-sum and it has no effect on its Nash equilibria.

Table 1: Property comparison of different preference optimization algorithms. The algorithms are compared based on whether they work for **general preferences**, whether they exhibit **last-iterate convergence** in two-player zero-sum games, and whether the output policy achieves **robust alignment**, i.e., a 50% win rate against other policies. $\not\checkmark$ : convergence only in the modified KL-regularized game $J_\tau(\pi_1, \pi_2, \pi_{\text{ref}})$ but not in $J(\pi_1, \pi_2)$.

| Algorithm | General Preference | Last-Iterate Convergence | Robust Alignment |
|---|---|---|---|
| DPO (Rafailov et al., 2024) | ✗ | ✗ | ✗ |
| IPO (Azar et al., 2024) | ✓ | ✗ | ✗ |
| SPPO (Wu et al., 2024) | ✓ | ✗ | ✗ |
| Nash-MD (Munos et al., 2024) | ✓ | $\not\checkmark$ | ✗ |
| INPO (Zhang et al., 2024) | ✓ | $\not\checkmark$ | ✗ |
| COMAL | ✓ | ✓ | ✓ |

## 3 A Convergent Meta-Algorithm for Alignment

We propose a simple meta-algorithm, **Co**nvergent **M**eta **Al**ignment **Al**gorithm (COMAL, Algorithm 1), for robustly aligning LLMs with general preferences. In Section 3.1 and 3.2, we present the theoretical foundations of COMAL and analyze its convergence properties. Section 3.3 describes its practical implementation that integrates COMAL with existing preference learning methods.

### 3.1 COMAL

COMAL (Algorithm 1) is an online iterative algorithm inspired by the classic conceptual prox-method (Nemirovski, 2004) first introduced in the optimization theory community. This method has recently been applied to finding a Nash equilibrium in zero-sum games (Perolat et al., 2021; Abe et al., 2024) and has had notable success in training advanced game AI models (Perolat et al., 2022).

---

**Algorithm 1: Co**nvergent **M**eta **Al**ignment **Al**gorithm (COMAL)

---

**Input:** Initial policy $\pi_{\text{sft}}$, preference oracle $\mathbb{P}$, regularization $\tau > 0$, number of iterations $T \geq 1$
**Output:** Optimized policy $\pi^T$
Initialize $\pi^1, \pi_{\text{ref}} \leftarrow \pi_{\text{sft}}$
**for** $t = 1, 2, \ldots, T - 1$ **do**
  $\pi^{t+1} \leftarrow \text{argmax}_{\pi_1} \min_{\pi_2} J_\tau(\pi_1, \pi_2, \pi_{\text{ref}})$ using Algorithm 2
  $\pi_{\text{ref}} \leftarrow \pi^{t+1}$
**return** $\pi^T$

---

**Algorithm 2:** Regularized game solver for $J_\tau(\pi_1, \pi_2, \pi_{\text{ref}}) - \text{argmax}_{\pi_1} \min_{\pi_2} J_\tau(\pi_1, \pi_2, \pi_{\text{ref}})$

---

**Input:** Reference policy $\pi_{\text{ref}}$, preference oracle $\mathbb{P}$, regularization $\tau > 0$, step size $\eta > 0$, number of iterations $K \geq 1$
**Output:** Regularized Nash equilibrium policy $\mu_K$
Initialize $\mu^1 \leftarrow \pi_{\text{ref}}$
**for** $k = 1, 2, \ldots, K - 1$ **do**
  $g_\tau^k \leftarrow \nabla_\mu(\mathbb{P}(\mu \succ \mu_k) - \tau \text{KL}(\mu || \pi_{\text{ref}})) = \mathbb{P}(\cdot \succ \mu_k) - \tau(\log \frac{\mu_k(\cdot)}{\pi_{\text{ref}}(\cdot)} + 1)$ // Gradient
  $\mu^{k+1} \leftarrow \text{Prox}(\mu_k, \eta g_\tau^k)$
**return** $\mu_K$

---

**Update Rule of COMAL**   In each iteration $t$, COMAL uses a regularized game solver (Algorithm 2) to update the next-iteration policy $\pi^{t+1}$ as the Nash equilibrium policy of a regularized game $J_\tau(\pi_1, \pi_2, \pi_{\text{ref}})$ using the current policy as reference $\pi_{\text{ref}} = \pi^t$. The rationale behind COMAL is simple: update the reference policy when no further progress can be made, which occurs when the algorithm reaches the Nash equilibrium of the regularized game. Denote $\pi^\star$ a Nash equilibrium of

the original game. We show that KL divergence to $\pi^\star$ is monotonically decreasing: $\mathrm{KL}(\pi^\star || \pi^{t+1}) \leq \mathrm{KL}(\pi^\star || \pi^t)$. Since $\pi^{t+1}$ is closer to the Nash equilibrium than $\pi^t$, COMAL updates the reference policy from $\pi^t$ to $\pi^{t+1}$ for further optimization. We also note that in COMAL, $\mathrm{KL}(\pi^\star || \pi^{t+1}) \leq \mathrm{KL}(\pi^\star || \pi^t)$ holds for any $\tau > 0$, allowing us to choose the regularization parameter $\tau_t > 0$ adaptively during the training process, without requiring it to decrease over time.

**Implementation of COMAL**   Each iteration of COMAL requires solving a zero-sum game with additional KL regularization $J_\tau(\pi_1, \pi_2, \pi_{\mathrm{ref}})$. We will show momentarily that many existing policy optimization methods for alignment can be applied to the KL regularized game and have exponentially fast convergence. We also present one practical implementation of COMAL integrated with INPO (Zhang et al., 2024) as the regularized game solver in Algorithm 4.

**Last-Iterate Convergence**   We prove that the meta-algorithm COMAL achieves last-iterate convergence to a Nash equilibrium, thereby ensuring robust alignment, which, to our knowledge, is the first result of its kind in the context of LLM alignment. The proof is in Appendix C.

**Theorem 1.** *We assume that there exists a Nash equilibrium $\pi^\star$ of $J(\pi_1, \pi_2)$ (defined in (3)) such that $\mathrm{supp}(\pi^\star) = \mathrm{supp}(\pi_{\mathrm{sft}})$. In every iteration $t \geq 1$, it holds that $\mathrm{KL}(\pi^\star || \pi^{t+1}) \leq \mathrm{KL}(\pi^\star || \pi^t)$. Moreover, COMAL has last-iterate convergence, i.e., $\lim_{t \to \infty} \pi^t$ exists and is a Nash equilibrium.*

### 3.2   Solving a Regularized Game

We present Mirror Descent (MD) in Algorithm 2 to compute a Nash equilibrium of the regularized game $J_\tau(\pi_1, \pi_2, \pi_{\mathrm{ref}})$. MD uses the prox operator as building blocks and we later show how to implement the prox operator using existing policy optimization algorithms. For simplicity, we consider policy $\pi \in \Delta(\mathcal{Y})$ and omit the dependence on the instruction $x$. All discussions can be extended to the contextual setting in a straightforward way.

**Mirror Descent and Multiplicative Weights Update**   Mirror Descent (MD) is a classical family of optimization algorithms (Nemirovskij and Yudin, 1983). An important member of this family is the Multiplicative Weights Update (MWU) algorithm (Arora et al., 2012), which is MD with negative entropy regularization. For a maximization problem $\max_\pi f(\pi)$, given an existing policy $\pi^t$, MWU computes the update $\pi^{t+1}$ as follows:

$$\pi^{t+1} := \arg\max_\pi \left\langle \nabla f(\pi^t), \pi \right\rangle - \eta^{-1} \cdot \mathrm{KL}(\pi || \pi^t). \tag{4}$$

Note that RLHF in (1) is equivalent to MWU if we interpret $f(\pi)$ as the expected reward under $\pi$ $\mathbb{E}_{y \sim \pi}[r(y)]$, and the gradient $\nabla f(\pi_{\mathrm{ref}})$ corresponds directly to $r$.

**Prox operator.**   The update rule of MWU can be succinctly expressed using the *prox operator* as shown in Algorithm 2.[3] Fix a 1-strongly convex function $\varphi : \mathcal{Z} \to \mathbb{R}$ over a closed convex set $\mathcal{Z} \subset \mathbb{R}^n$. The *Bregman divergence* induced by $\varphi$ is

$$D_\varphi(\cdot || \cdot) : \mathcal{Z} \times \mathcal{Z} \to \mathbb{R}_{\geq 0},$$
$$D_\varphi(z || z') := \varphi(z) - \varphi(z') - \langle \nabla \varphi(z'), z - z' \rangle.$$

Given a reference point $z \in \mathcal{Z}$ and a vector $g \in \mathbb{R}^n$, the prox operator $\mathrm{Prox}(z, g)$ generalizes the notion of a gradient ascent step from $z$ in the direction of $g$.

**Definition 3** (Prox Operator). *For a strongly convex regularizer $\varphi$, the prox operator is defined as*

$$\mathrm{Prox}(z, g) := \arg\max_{z'} \langle g, z' \rangle - D_\varphi(z' || z) = \arg\max_{z'} \langle g + \nabla \varphi(z), z' \rangle - \varphi(z').$$

When $\varphi(z) = \frac{1}{2} \|z\|_2^2$ is the $\ell_2$ regularizer, the prox operator $\mathrm{Prox}(z, g) = \Pi_\mathcal{Z}[z + g]$ is the exactly the projected gradient ascent step. In this paper, without additional notes, we choose $\varphi = \sum_{i=1}^n z[i] \ln z[i]$ as the negative entropy regularizer and the corresponding Bregman divergence $D_\varphi$ is the KL divergence. The update rule of MWU in (4) is equivalent to $\pi^{t+1} = \mathrm{Prox}(\pi^t, \eta \nabla f(\pi^t))$

---

[3]The prox operator is also called the prox-mapping (Nemirovski, 2004).

**Exponentially Fast Convergence**    Denote $\pi_\tau^\star$ the Nash equilibrium of the KL regularized game $J_\tau(\pi_1, \pi_2, \pi_{\text{ref}})$, which is $\tau$-strongly monotone. We can apply existing results (Abe et al., 2024) to show that MWU (Algorithm 2) achieves linear last-iterate convergence rate: the KL divergence to the Nash equilibrium $\pi_\tau^\star$ decreases exponentially fast. The proof is in Appendix D.

**Theorem 2.** *For appropriate step size $\eta > 0$, Algorithm 2 guarantees for every $k \geq 1$,* $\text{KL}(\pi_\tau^\star || \mu^{k+1}) \leq (1 - \frac{\eta\tau}{2})^k \text{KL}(\pi_\tau^\star || \pi_{\text{ref}})$.

## 3.3 PRACTICAL METHODS FOR COMPUTING THE PROX OPERATOR

We show how to implement COMAL in practical large-scale applications like LLM alignment by computing the prox operator. Specifically, we observe that many existing algorithms designed for RLHF and preference optimization with neural network parameters can be adapted to solve the prox operator $\text{Prox}(\pi, \eta g)$ ($\eta > 0$ is the step size). These algorithms include RL algorithms like PPO and loss-minimization algorithms like DPO, IPO, SPPO, DRO, INPO, each of which may be preferred in certain settings. Due to space limit, we only present IPO and INPO here but defer discussion of other methods to Appendix E. Our contribution here is not proposing new algorithms but unifying existing diverse preference methods through the perspective of computing the prox operator. This perspective opens the possibility of applying other algorithms from online learning and optimization to robust LLM alignment and we include implementation for two other algorithms in Appendix F.

**IPO for computing** $\text{Prox}$ **for general preferences**    We assume a general preference model $\mathbb{P}$ over $\mathcal{Y}$ (not necessarily the BT model). We consider the case where $g$ is the win-rate against some policy $\mu$ such that $g_\mu(y) = \mathbb{P}[y \succ \mu] := \mathbb{E}_{y' \sim \mu}[\mathbb{P}[y \succ y']]$ (think of $\mu$ as the reference policy $\pi_{\text{ref}}$ or other online policy $\pi_t$). We assume the dataset contains win-lose pairs sampled from $\mu$: $\{y_w, y_l \sim \mu\}$. We denote the preference distribution $\lambda_\mathbb{P}(y, y')$ as a binary distribution:

$$\lambda_\mathbb{P}(y, y') = \begin{cases} (y, y') \text{ with probability } \mathbb{P}[y \succ y'] \\ (y', y) \text{ with probability } 1 - \mathbb{P}[y \succ y'] \end{cases} \tag{5}$$

The (population) IPO loss (Azar et al., 2024) is defined as

$$\ell_{\text{IPO}}(\theta, \mu) := \mathbb{E}_{(y_w, y_l) \sim \mu, (y^+, y^-) \sim \lambda_\mathbb{P}(y_w, y_l)} \left[ \left( \log \frac{\pi_\theta(y^+)}{\pi(y^+)} - \log \frac{\pi_\theta(y^-)}{\pi(y^-)} - \frac{\eta}{2} \right)^2 \right].$$

Azar et al. (2024) have shown that the minimizer of the $\ell_{\text{IPO}}(\theta, \mu)$ satisfies

$$\pi_\theta(y) \propto \pi(y) \exp\left(-\eta \mathbb{P}[y \succ \mu]\right) \Leftrightarrow \pi_\theta = \text{Prox}(\pi, \eta g_\mu).$$

Thus we can compute the prox operator $\text{Prox}(\pi, \eta g_\mu)$ where $g_\mu = \mathbb{P}(\cdot \succ \mu)$ by minimizing the IPO loss against policy $\mu$.

**INPO for computing** $\text{Prox}$ **for regularized preferences**    A generalization of the IPO loss to the regularized preference setting is the Iterative Nash Policy Optimization (INPO) loss (Zhang et al., 2024). Here, we define $g_\mu^\tau$ the gradient $\nabla_\pi J_\tau(\pi, \mu, \pi_{\text{ref}}) = \mathbb{P}(\cdot \succ \mu) - \tau \log \frac{\mu(\cdot)}{\pi_{\text{ref}}(\cdot)}$ of the regularized objective. The INPO algorithm with the corresponding INPO loss is shown in Algorithm 3. Similarly, it has been shown that the INPO loss minimizer corresponds to the prox operator's solution $\text{Prox}(\pi, \eta g_\mu^\tau)$ (Zhang et al., 2024). Thus we can apply INPO in Algorithm 2 directly.

**Practical Implementation of COMAL**    We present an implementation of COMAL in Algorithm 4 using the INPO (Zhang et al., 2024) as a subgame solver. We remark that COMAL can also be implemented using PPO or many other preference learning algorithms, as we show in Section 3.3 and Appendix E. Given the implementation of these existing methods, our meta-algorithm requires minimal change but archives last-iterate convergence to a Nash equilibrium.

## 4 SYNTHETIC EXPERIMENTS

We conduct experiments on a simple bandit problem with $\mathcal{Y} = \{y_a, y_b, y_c\}$ and non-BT preference model over $\mathcal{Y}$. Specifically, we set $\mathbb{P}[y_b \succ y_a] = \mathbb{P}[y_c \succ y_b] = 0.9$ and $\mathbb{P}[y_a \succ y_c] = 0.8$. Observe that the preference is intransitive and exhibits a preference cycle $y_c \succ y_b \succ y_a \succ y_c$. The setup for the synthetic experiment is included in Appendix G.

---

**Algorithm 3:** INPO (Zhang et al., 2024) for solving $J_\tau(\pi_1, \pi_2, \pi_{\text{ref}})$

---

**Input:** Reference policy $\pi_{\text{ref}}$, regularization $\tau > 0$, step size $\eta > 0$, number of rounds $K \geq 1$, preference oracle $\mathbb{P}$.

**Output:** Approximate regularized Nash equilibrium policy $\mu_K$

Initialize $\mu^1 \leftarrow \pi_{\text{ref}}$

**for** $k = 1, 2, \ldots, K - 1$ **do**

    Generate response pairs $\{y_1^{(i)}, y_2^{(i)}\}_{i=1}^n$ where $y_1^{(i)}, y_2^{(i)} \sim \mu^k$

    Query preference oracle $\mathbb{P}$ to get preference data $\mathcal{D}_k = \{y_w^{(i)}, y_l^{(i)}\}_{i=1}^n$

    Compute $\mu^{k+1} = \arg\min_{\pi \in \Pi} \mathbb{E}_{(y_w, y_l) \sim \mathcal{D}_k} \ell_{\text{INPO}}(\pi)$ where

$$\ell_{\text{INPO}}(\pi) := \mathbb{E}_{\underset{(5)}{(y^+, y^-) \sim \lambda_{\mathbb{P}}(y_w, y_l)}} \left[ \left( \log \frac{\pi(y^+)}{\pi(y^-)} - \eta\tau \log \frac{\pi_{\text{ref}}(y^+)}{\pi_{\text{ref}}(y^-)} - (1 - \eta\tau) \log \frac{\mu^t(y^+)}{\mu^t(y^-)} - \frac{\eta}{2} \right)^2 \right]$$

**return** $\mu^K$

---

**Algorithm 4:** Practical Implementation of COMAL integrated with INPO (Algorithm 3)

---

**Input:** Initial policy $\pi_{\text{sft}}$, regularization $\{\tau_t > 0\}$, step size $\{\eta_t > 0\}$, number of iterations $T \geq 1$, number of inner optimization steps $\{K_t \geq 1\}$, preference oracle $\mathbb{P}$.

**Output:** Optimized policy $\pi^T$

Initialize $\pi^1, \pi_{\text{ref}} \leftarrow \pi_{\text{sft}}$

**for** $t = 1, 2, \ldots, T - 1$ **do**

    $\pi^{t+1} \leftarrow \text{INPO}(\pi_{\text{ref}}, \tau_t, \eta_t, K_t, \mathbb{P})$ defined in Algorithm 3

    $\pi_{\text{ref}} \leftarrow \pi^{t+1}$

**return** $\pi^T$

---

**Experiments using noiseless gradient** We present numerical results of mirror-descent (MD) algorithms (equivalent to MWU) and COMAL (Algorithm 1) in Figure 1. We can see that the MD algorithm diverges from the unique Nash equilibrium and suffers a large equilibrium gap, while COMAL achieves fast last-iterate convergence to the Nash equilibrium, aligned with our theoretical results (Theorem 1).

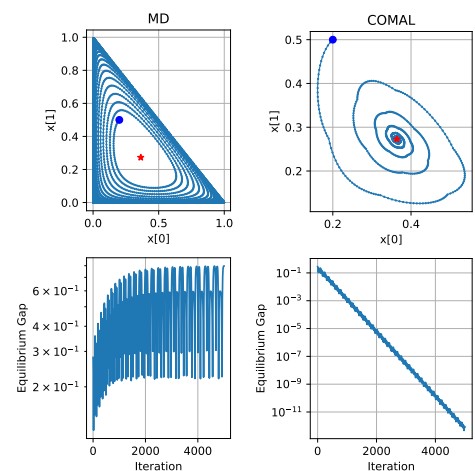

Figure 1: Dyanmics on a simple 3-dimensional preference game. The unique Nash equilibrium is $[4/11, 3/11, 4/11]$ represented as red star. We initialize all algorithms at the blue dot point $[0.2, 0.5, 0.3]$.

**Experiements using preference samples** Since the popular iterative DPO algorithm does not contain a gradient step, we also conduct experiments with only Oracle query access to the preference model. We compare the performance of various algorithms, including iterative DPO, iterative IPO, SPPO, and INPO and present results in Figure 2. The sample-only setting is also more aligned with what happens in practice. We use a sufficient number of samples in each iteration for every algorithm. As a result, the COMAL performs the same as in the noiseless gradient setting, while the iterative IPO algorithm becomes equivalent to the MD algorithm. We note the following:

- Iterative DPO: We observe that iterative DPO diverges and cycles between extreme policies (e.g., outputting $y_a$ with probability close to 1). This is aligned with (Azar et al., 2024), where they found DPO will converge to the deterministic policy regardless of the regularization parameter in extreme preference settings. The cycling behavior of iterative DPO may be explained as follows: in each iteration, DPO converges to a nearly deterministic policy output $y$; then the new preference

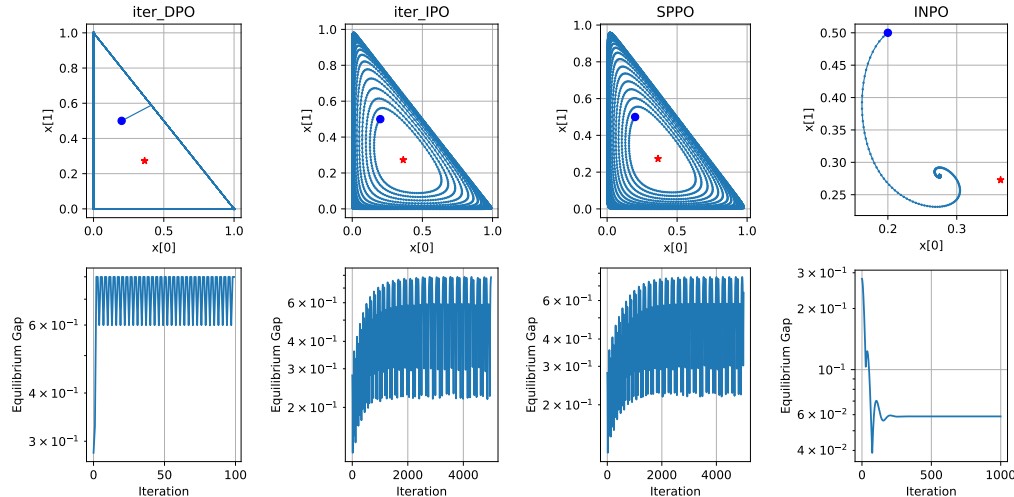

Figure 2: Dyanmics on a simple 3-dimensional preference game. The unique Nash equilibrium is $[4/11, 3/11, 4/11]$ represented as red star. We initialize all algorithms at the blue dot point $[0.2, 0.5, 0.3]$.

data shows that $y' \neq y$ is more preferred; finally, iterative DPO cycles over $\mathcal{Y}$ since the preference itself exhibits a cycle and there is no clear winner.

- Iterative IPO (Azar et al., 2024; Calandriello et al., 2024): The IPO loss is a variant of the DPO loss, but it does not rely on the BT model assumption and works for a general preference model. However, as we have discussed before, (exactly) minimizing the IPO loss is equivalent to performing one MD step, and thus, iterative IPO is equivalent to MD up to sampling error. As a result, we observe that iterative IPO also exhibits cycling behavior.

- SPPO (Wu et al., 2024): The SPPO algorithm (see Appendix E) is not exactly the same as MWU since SPPO assumes the partition function is always $Z = \log \frac{\eta}{2}$ which may not be the case. We observe that SPPO exhibits very similar cycling behavior as MD. We conclude that SPPO approximates MD very well in this instance and exhibits similar behavior.

- INPO (Zhang et al., 2024): The INPO algorithm is designed for finding the Nash equilibrium of the KL regularized game $J_\tau(\pi_1, \pi_2, \pi_{\text{ref}})$. As we proved in Theorem 2, INPO does not diverge and exhibits last-iterate convergence. However, it converges to a point that differs from the Nash equilibrium of the game $J(\pi_1, \pi_2)$ and, as a result, lacks the robust alignment property.

## 5 REAL-WORLD EXPERIMENTS

Apart from the controlled synthetic experiments, we conduct experiments with a pre-trained LLM, Qwen2-1.5B (Yang et al., 2024), on a commonly used dataset UltraFeedback (Cui et al., 2023) to show the effectiveness of COMAL under the real-world preference optimization setting.

### 5.1 EXPERIMENTAL SETTINGS

**Datasets** We use the UltraFeedback dataset, specifically its binarized version for preference fine-tuning.[4] It contains 64K data examples consisting of a user instruction and a positive-negative output pair annotated by GPT-4. The instructions in this dataset span a wide range of types, making it well-suited for studying preference optimization in real-world settings. Since we focus on online and iterative preference optimization, only the instructions are used because the output pairs will be generated and annotated online. In addition, to reduce the computational cost, the instructions are randomly split into 6 equal-size subsets. Each subset therefore contains around 10K instructions and is used in one training iteration.

---

[4]https://huggingface.co/datasets/HuggingFaceH4/ultrafeedback_binarized.

**Preference Oracle** The preference oracle we used is Llama-3-OffsetBias-8B (Park et al., 2024a), which is a pairwise preference model that predicts which output is better given an instruction and a pair of outputs. Fine-tuned from Meta-Llama-3-8B-Instruct (Dubey et al., 2024), it achieves strong performance on various human preference alignment benchmarks in RewardBench (Lambert et al., 2024). We selected it as the preference oracle for its balance of computational efficiency and alignment with human preferences, making it suitable for iterative preference optimization.

**Online Preference Data Generation** To construct the preference data, i.e., output pairs with a preference annotation specifying which one is better, we adopt the setting of Zhang et al. (2024) by sampling 5 candidate outputs for each instruction with a temperature of 0.8 and applying the preference oracle to compare all the output pairs constructed. The best and the worst candidate outputs, derived from the pairwise comparison results, are then selected to form a data point.

**Baselines** We include the following baselines for comparisons with COMAL: (1) SFT, which fine-tunes the pre-trained Qwen2-1.5B on the UltraChat dataset, with the resulting checkpoint serving as the starting point and/or reference policy for the other training algorithms; (2) vanilla DPO (Rafailov et al., 2024) and (3) vanilla IPO (Azar et al., 2024), where one training iteration is performed over the entire instruction set of UltraFeedback with output pairs sampled from the SFT policy; (4) INPO (Zhang et al., 2024), where each iteration of training is performed on a single data split; (5) iterative IPO, which follows a training setting similar to INPO but without the KL regularization with respect to the reference policy.

**Evaluations** We use the instructions in a widely used benchmark, AlpacaEval (Li et al., 2023), to construct the test set, since these instructions are diverse and cover various task scenarios. However, instead of using GPT-4, the default evaluator for the AlpacaEval benchmark, we chose to use the same preference oracle used during data generation, Llama-3-OffsetBias-8B, as the evaluator. This decision was made to maintain a controlled experimental setting, ensuring that the preference oracle the model learns to fit is also the one used to evaluate its performance.

**Training Details** We follow the training recipe proposed in Tunstall et al. (2023) for the experiments. Specifically, at each training iteration, the models are fine-tuned for 3 epochs with a batch size of 32 and maximum learning rate of $5 \times 10^{-7}$, using a linear learning rate scheduler where 10% of the steps are for warmup and the rest for linearly decreasing the rate. The checkpoints are selected based on their validation loss on the UltraFeedback dataset. The training is performed on 8 NVIDIA A6000 Ada GPUs with 48GB memory, and one training iteration over the 10K instructions takes around 5 hours. Due to the relatively high computational requirements and the large number of training iterations we tested (up to 18), we opted to use a moderately sized LLM and did not conduct an exhaustive hyper-parameter search, instead referencing settings from previous work when appropriate. To the best of our knowledge, multi-iteration training like ours has rarely been explored in previous work. For example, INPO (Zhang et al., 2024) only performed optimization for up to 3 iterations, which is equivalent to just one full round over UltraFeedback's instructions.

**Hyper-Parameters** We conduct a grid search for the strength of the KL regularization, $\eta^{-1}$, in both vanilla DPO and IPO. We found that DPO achieves the best performance when $\eta^{-1}$ is set to 0.01, while IPO achieves the best performance when $\eta^{-1}$ is set within the range of 0.002 - 0.01. We then choose the value of $\eta^{-1}$ to be 0.002 to encourage larger learning steps.[5] This value of $\eta$ is also used for iterative IPO and INPO. INPO has another hyper-parameter $\tau$ which controls the strength of the KL regularization from the reference policy. We determine its value following the setting of Zhang et al. (2024), where $\eta\tau$ is set to a fixed ratio, $1/3$. Regarding COMAL, which is implemented based on INPO as outlined in Algorithm 4, the reference policy is updated when the first optimization step begins to converge or overfit, and $\eta^{-1}$ is increased to 0.01 to improve training stability.

## 5.2 RESULT ANALYSIS

Figure 3 presents the training dynamics of three iterative preference optimization algorithms we compared: iterative IPO (Iter-IPO), INPO, and COMAL, which are demonstrated by their checkpoints' win rates against the SFT checkpoint and the average length of their outputs. For INPO and COMAL, the model is trained for up to 18 iterations, which are equivalent to 3 training rounds over the entire instruction set since it has been split into 6 subsets. We note that:

---

[5]More details are in Appendix H.

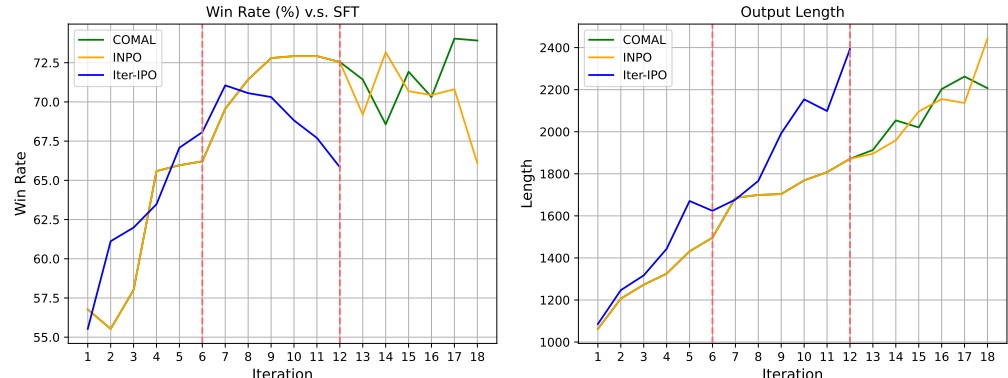

Figure 3: Comparisons of Iterative IPO (Iter-IPO), INPO, and COMAL. The win rate of the trained checkpoints against the SFT checkpoint, and the average length of the outputs are compared. The red vertical lines mark the end of one training round – a full pass over the 6 instruction splits.

Table 2: Performance comparison of different training algorithms. The row v.s. column win rate (%) is reported. For INPO, we report its performance with 2-round (R2) and 3-round (R3) training.

| Row/Col | SFT | DPO | IPO | Iter-IPO | INPO-R2 | INPO-R3 | COMAL | Avg |
|---|---|---|---|---|---|---|---|---|
| Iter-IPO | 65.84 | 56.40 | 54.04 | 50.00 | 47.83 | 46.21 | 39.01 | 51.33 |
| INPO-R2 | 72.55 | 60.25 | 58.39 | 52.17 | 50.00 | 49.32 | 41.37 | 54.86 |
| INPO-R3 | 66.09 | 60.25 | 58.51 | 53.79 | 50.68 | 50.00 | 44.97 | 54.90 |
| COMAL | **73.91** | **66.71** | **66.21** | **60.99** | **58.63** | **55.03** | **50.00** | **61.64** |

(1) Iter-IPO shows a quicker improvement rate at the beginning of the training, but its performance against the SFT checkpoint starts to degrade in the second training round, which indicates the inherent instability of this training algorithm.

(2) INPO archives a relatively stable win rate against SFT at the end of the second training round. However, its win rate starts to slightly degrade in the third training round. We suspect this suggests that INPO has started to converge and/or overfit. Therefore, for COMAL, which shares the same training trajectory as INPO for the first two training rounds, we update the reference policy at the beginning of the third training round, following the optimization process described in Algorithm 4.

(3) COMAL is able to further improve the model performance with the updated reference policy. Notably, it also results in the shortest outputs, suggesting that it is more robust to the length bias of the preference models which preference optimization algorithms tend to exploit (Park et al., 2024b).

Table 2 provides pairwise comparisons between the final checkpoints of the iterative preference optimization algorithms and a few baselines. It demonstrates the clear advantage of COMAL, which is able to achieve an above 50% win rate against all the other checkpoints. In contrast, Iter-IPO can only outperform the vanilla DPO and IPO settings. Regarding INPO, we found that the average win rate of its checkpoint after the third training round (INPO-R3) is only slightly higher than that of its intermediate checkpoint at the end of the second training round (INPO-R2) (54.90 vs. 54.86), suggesting that its performance plateaued by the end of the second training round. Conversely, by updating the reference policy, COMAL further improves the performance in the third training round. In Appendix I, we provide additional evaluation results such as using GPT-4 as the preference oracle, which further demonstrates the effectiveness of COMAL compared to other algorithms.

## 6 CONCLUSION

We have proposed COMAL, a meta-algorithm for preference optimization that provably converges to the Nash equilibrium policy in the last iterate. We have provided a theoretical analysis of the properties of COMAL and have empirically demonstrated its effectiveness under both synthetic and real-world experimental settings. We believe COMAL has significant potential to enhance the performance of LLMs in the alignment fine-tuning setting, due to its theoretical guarantees and flexibility, as it can be integrated with existing learning algorithms while overcoming their limitations.

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

## CONTENTS

## A  RELATED WORK

**Alignment under Preference models**   Most existing approaches adopt the Bradley-Terry (BT) preference model (Bradley and Terry, 1952; Christiano et al., 2017), which involves first learning a preference model and then optimizing the objective function with a KL divergence penalty relative to the original language model. For example, RLHF (Ouyang et al., 2022) aims to ensure that LLMs follow instructions by initially learning a BT model and subsequently fine-tuning the model based on the learned reward while regularizing it with the original LLM.

Building on this framework, Rafailov et al. (2024) introduces Direct Preference Optimization (DPO) that maintains the assumption of the BT model for preferences but eliminates the preference learning step by reformulating the objective and optimizing it directly. Additionally, Ethayarajh et al. (2024) diverges from the traditional BT-based methods by deriving algorithms that bypass the preference modeling step altogether. Instead, they model user preferences based on Kahneman and Tversky's utility theory.

**Alignment Solution Concepts under General Preferences**  Azar et al. (2024) are the first to consider general preferences and propose a family of optimization objectives that optimize a function of the preferences probabilities regularized by the KL divergence with respect to the original model. They propose the IPO algorithm, an offline algorithm that directly optimizes the win rate of the model penalized by the KL divergence with respect to the original model. Munos et al. (2024) also consider general preferences and aim to find the *von Neumann winner*, which corresponds to the Nash equilibrium of a game played between the two LLMs over the win rate. They propose a variant of the Mirror Descent (MD) algorithm called Nash-MD and show last-iterate convergence in the KL-regularized game. Concurrently, Swamy et al. (2024) study the same solution concept focusing more on sequential games. Calandriello et al. (2024) proved that the objective of the the IPO algorithm coincides with the Nash policy under a proper choice of the parameter that controls the regularization.

**Iterative Self-Play Algorithms**  Apart from the aforementioned works, a line of recent work also propose practical implementation of the Mirror Dscent (MD) algorithms, which can be used to learn the Nash equilibrium via self-play. Rosset et al. (2024) propose Direct Nash Optimization (DNO), where at each iteration, the model regresses the predicted preferences against the actual preferences using cross-entropy loss. Similarly, Wu et al. (2024) introduce the Self-Play Preference Optimization (SPPO) method, Gao et al. (2024) introduce Reinforcement Learning via Regressing Relative Rewards (REBEL), and Richemond et al. (2024) introduce the Direct Reward Optimization (DRO) which regresses the loss using the $L_2$ distance at each iteration. Since these algorithms simulate the MD update, when applied in a (unregularized) zero-sum game, they only have average-iterate convergence but all *diverge in last iterate*. Moreover, all these methods require the estimation of the win rate, which can be computationally intensive and may introduce estimation errors.

Most closely related to our work is Iterative Nash Policy Optimization (INPO) by Zhang et al. (2024), which continues to use $L_2$ distance regression. However, by further reformulating and simplifying the objective similar to IPO, INPO eliminates the need to estimate the expected win rate. The primary distinction between our approach and INPO is that INPO is designed for the KL-regularized game and is equivalent to MD; while our algorithm COMAL is inspired by the Conceptual Prox algorithm and guarantees last-iterate convergence in the unregularized game. This fundamental difference allows COMAL to achieve more favourable convergence properties with robust alignment (i.e., 50% against any other policy) for large language models.

**Last-Iterate Convergence on Games**  It is well-established that Mirror Descent fails to converge in simple zero-sum games, often resulting in cycling behavior (Mertikopoulos et al., 2018). In contrast, several prominent algorithms have been shown to achieve last-iterate convergence including the Proximal Point (PP) method (Rockafellar, 1976), Extra-Gradient (EG) (Korpelevich, 1976), Optimistic Online Mirror Dscent (OOMD) (Popov, 1980; Rakhlin and Sridharan, 2013), and the Conceptual Prox/Mirror Prox methods (Nemirovski, 2004), in the more general setting of monotone variational inequality that covers zero-sum games as special cases. The asymptotic convergence properties of these algorithms have been extensively studied (Popov, 1980; Facchinei and Pang, 2003; Iusem et al., 2003; Nemirovski, 2004; Daskalakis and Panageas, 2018). In particular, variants of the Conceptual Prox algorithm (Nemirovski, 2004) instantiated with different regularizers have been shown to have last-iterate convergence in zero-sum matrix games (Abe et al., 2024) and zero-sum imperfect-information games (Perolat et al., 2021). Recently, there has been a growing focus on establishing finite-time convergence guarantees for these methods, addressing the practical necessity of understanding their performance within a limited number of iterations (see e.g. (Mokhtari et al., 2020b;a; Golowich et al., 2020b;a; Bauschke et al., 2021; Wei et al., 2021; Cai et al., 2022; Gorbunov et al., 2022) and references therein). Although last-iterate convergent algorithms have been extensively studied in optimization and game theory, only the divergent MD algorithm has been implemented for LLM alignment (see the "Iterative Self-Play Algorithms" paragraph above). Compared to the

works mentioned above, our work is the first to (1) introduce last-iterate convergent methods to LLM alignment; (2) provide the first practical implementation of these methods in the context of LLM fine-tuning, including the Conceptual Prox method, the Mirror-Prox (MP) method, and the Optimistic Multiplicative Weights Update (OMWU) algorithm (see Appendix F for details for MP and OMWU); (3) conduct LLM-based experiments and demonstrate the empirical success of the conceptual prox method in large-scale experiments. Our work demonstrates the value of applying theoretical insights from optimization and game theory to LLM alignment in practice.

## B  PROPERTIES OF THE PROX OPERATOR

Recall that $\text{Prox}(z, g) = \text{argmax}_{z' \in \mathcal{Z}} \langle g, z' \rangle - D_\varphi(z'||z) = \text{argmax}_{z' \in \mathcal{Z}} \langle g + \nabla\varphi(z), z' \rangle - \varphi(z')$. The following properties of the prox operator are well-known in the literature(e.g., (Nemirovski, 2004))

**Lemma 1.** $\text{Prox}(z, g) = z'$ if and only if $\langle g + \nabla\varphi(z) - \nabla\varphi(z'), z' - z^* \rangle \geq 0$ for all $z^* \in \mathcal{Z}$.

**Corollary 1.** Let $\text{Prox}(z, g) = z'$, then

$$\langle g, z^* - z' \rangle \leq D_\varphi(z^*||z) - D_\varphi(z^*||z') - D_\varphi(z'||z), \quad \forall z^* \in \mathcal{Z}$$

## C  PROOF OF THEOREM 1

The proof of Theorem 1 is relatively standard in the literature (Facchinei and Pang, 2003; Nemirovski, 2004). We include a formal proof here for completeness. In the proof, we assume that each step of COMAL, $\pi^{t+1} \leftarrow \text{argmax}_{\pi_1} \min_{\pi_2} J_\tau(\pi_1, \pi_2, \pi_{\text{ref}})$ can be solved exactly. Our proof extends to the case the optimization problem is solved approximately with sufficient accuracy.

In Theorem 1, we make the following assumption.

**Assumption 1.** *We assume there exists a Nash equilibrium $\pi^\star$ such that $\text{supp}(\pi^\star) = \text{supp}(\pi_{\text{sft}})$.*

This assumption is mild and **much weaker** than the "Bounded Log Density" assumptions used in previous works (Rosset et al., 2024; Zhang et al., 2024), which requires $|\log \frac{\pi^t}{\pi_{\text{sft}}}|$ is bounded.

Recall that $\Pi := \{\pi : \text{supp}(\pi) \subseteq \text{supp}(\pi_{\text{sft}})\}$. Then $\text{KL}(\pi||\pi_{\text{sft}}) \leq D := \max_{y:\pi_{\text{sft}}(y)>0} \log \pi_{\text{sft}}(y)$ is bounded for any $\pi \in \Pi$. We first prove $\text{KL}(\pi^\star||\pi^{t+1}) \leq \text{KL}(\pi^\star||\pi^t)$ for any $t \geq 1$.

**Lemma 2.** *Let $\pi^\star$ be an Nash equilibrium of $J(\pi_1, \pi_2)$. Then for any $\tau > 0$, if*

$$(\pi, \pi) = \underset{\pi_1 \in \Pi}{\text{argmax}} \underset{\pi_2 \in \Pi}{\text{argmin}} J_\tau(\pi_1, \pi_2, \pi_{\text{ref}}),$$

*then*

$$\text{KL}(\pi^\star||\pi) \leq \text{KL}(\pi^\star||\pi_{\text{ref}}) - \text{KL}(\pi||\pi_{\text{ref}})$$

*Proof.* By definition of the prox operator, we have

$$\pi = \underset{\pi_1 \in \Pi}{\text{argmax}} J_\tau(\pi_1, \pi, \pi_{\text{ref}})$$

$$= \underset{\pi_1 \in \Pi}{\text{argmax}} \mathbb{P}(\pi_1 \succ \pi) - \tau \text{KL}(\pi_1, \pi_{\text{ref}})$$

$$= \text{Prox}(\pi_{\text{ref}}, \frac{1}{\tau}\mathbb{P}(\cdot \succ \pi)). \tag{6}$$

Using Corollary 1, we have for any $\pi' \in \Pi$,

$$\frac{1}{\tau}(\mathbb{P}(\pi' \succ \pi) - \mathbb{P}(\pi \succ \pi)) \leq \text{KL}(\pi'||\pi_{\text{ref}}) - \text{KL}(\pi'||\pi) - \text{KL}(\pi||\pi_{\text{ref}}). \tag{7}$$

Plugging $\pi' = \pi^\star$ into the above inequality and noting that $\mathbb{P}(\pi \succ \pi) = \frac{1}{2}$, we get

$$\frac{1}{\tau}\left(\mathbb{P}(\pi^\star \succ \pi) - \frac{1}{2}\right) \leq \text{KL}(\pi^\star||\pi_{\text{ref}}) - \text{KL}(\pi^\star||\pi) - \text{KL}(\pi||\pi_{\text{ref}}).$$

Since $\pi^\star$ is a Nash equilibrium and thus $\mathbb{P}(\pi^\star \succ \pi) \geq \frac{1}{2}$, the lefthand side of the above inequality is $\geq 0$. The we have

$$\mathrm{KL}(\pi^\star || \pi) \leq \mathrm{KL}(\pi^\star || \pi_{\mathrm{ref}}) - \mathrm{KL}(\pi || \pi_{\mathrm{ref}}).$$

$\square$

Lemma 2 implies the following properties on the trajectory $\{\pi^t\}$.

**Corollary 2.** *In COMAL, we have*

1. $\mathrm{KL}(\pi^\star || \pi^{t+1}) \leq \mathrm{KL}(\pi^\star || \pi^t)$ *for all* $t \geq 1$.

2. $\sum_{t=1}^\infty \mathrm{KL}(\pi^{t+1} || \pi^t) \leq \mathrm{KL}(\pi^\star || \pi_{\mathrm{sft}}) < +\infty$.

3. $\mathrm{supp}(\pi^t) = \mathrm{supp}(\pi_{\mathrm{sft}})$ *for all* $t \geq 1$.

*Proof.* The first item is direct from Lemma 2. The second item is also direct by applying Lemma 2 for $t = 1, 2 \dots$:

$$\sum_{t=1}^\infty \mathrm{KL}(\pi^{t+1} || \pi^t) \leq \sum_{t=1}^\infty \mathrm{KL}(\pi^\star || \pi^t) - \mathrm{KL}(\pi^\star || \pi^{t+1}) \leq \mathrm{KL}(\pi^\star || \pi_{\mathrm{sft}}) \leq D < \infty.$$

For the third item, let $\pi^\star$ be a Nash equilibrium such that $\mathrm{supp}(\pi^\star) = \mathrm{supp}(\pi_{\mathrm{sft}})$ as guaranteed by Assumption 1. On one hand, since $\mathrm{KL}(\pi^t || \pi^{t-1}) < \infty$ for all $t \geq 1$, we have $\mathrm{supp}(\pi^t) \subseteq \mathrm{supp}(\pi^{t-1}) \subseteq \dots \subseteq \mathrm{supp}(\pi_{\mathrm{sft}})$. On the other hand, $\mathrm{KL}(\pi^\star || \pi^t) < \infty$ implies $\mathrm{supp}(\pi^\star) \subseteq \mathrm{supp}(\pi^t)$. Since $\mathrm{supp}(\pi_{\mathrm{sft}}) = \mathrm{supp}(\pi^\star)$, we have $\mathrm{supp}(\pi^t) = \mathrm{supp}(\pi_{\mathrm{sft}}) = \mathrm{supp}(\pi^\star)$. $\square$

Since the sequence $\{\pi^t\}$ is bounded (all lies in the simplex), it has at least one limit point $\hat{\pi}$. The next lemma shows that a limit point must be a Nash equilibrium.

**Lemma 3.** *If $\hat{\pi}$ is a limit point of $\{\pi^t\}$, then $\hat{\pi}$ is a Nash equilibrium of $J(\pi_1, \pi_2)$.*

*Proof.* By item 2 in Corollary 2, we have $\lim_{t \to \infty} \mathrm{KL}(\pi^{t+1} || \pi^t) = 0$. This implies $\lim_{t \to \infty} ||\pi^{t+1} - \pi^t|| = 0$. As $\hat{\pi}$ is a limit point of $\{\pi^t\}$, we let $\{\pi^k : k \in \kappa\}$ be the subsequence that converges to $\hat{\pi}$. Then by Equation (6), we have

$$\lim_{k \in \kappa, k \to \infty} \pi^{k+1} = \lim_{k \in \kappa, k \to \infty} \mathrm{Prox}(\pi^k, \frac{1}{\tau} \mathbb{P}(\cdot \succ \pi^{k+1}))$$

$$\Rightarrow \hat{\pi} = \mathrm{Prox}(\hat{\pi}, \frac{1}{\tau} \mathbb{P}(\cdot \succ \hat{\pi})).$$

Thus $\hat{\pi}$ is a fixed point of $\mathrm{Prox}(\pi, \frac{1}{\tau} \mathbb{P}(\cdot \succ \pi)$. Moreover, by item 3 in Corollary 2, we have $\mathrm{supp}(\hat{\pi}) = \mathrm{supp}(\pi_{\mathrm{sft}})$. Now consider both the max and min player running MWU initialized with $\pi^1 = \hat{\pi}$. Then we have $\pi^t = \hat{\pi}$ for all $t \geq 1$. By Equation (7), we have for any $\pi' \in \Pi$,

$$\frac{1}{\tau} \sum_{t=1}^\infty \left( \mathbb{P}(\pi' \succ \hat{\pi}) - \frac{1}{2} \right) \leq \mathrm{KL}(\pi' || \hat{\pi}) < \infty,$$

where the inequality holds since $\mathrm{supp}(\pi') \subseteq \mathrm{supp}(\hat{\pi})$. As a result, we get

$$\mathbb{P}(\pi' \succ \hat{\pi}) \leq \frac{1}{2}, \forall \pi' \in \Pi \Leftrightarrow \mathbb{P}(\hat{\pi} \succ \pi') \geq \frac{1}{2}, \forall \pi' \in \Pi$$

Thus $\hat{\pi}$ is a Nash equilibrium of $J(\pi_1, \pi_2)$. $\square$

**Proof of Theorem 1** Since $\hat{\pi}$ is a Nash equilibrium, by Corollary 2, $\{\mathrm{KL}(\hat{\pi} || \pi^t) \geq 0\}$ is a decreasing sequence. Thus $\{\mathrm{KL}(\hat{\pi} || \pi^t)\}$ converges. Let $\{\pi^k : k \in \kappa\}$ be a subsequence that converges to $\hat{\pi}$. Then we have

$$\lim_{t \to \infty} \mathrm{KL}(\hat{\pi} || \pi^t) = \lim_{k \in \kappa, k \to \infty} \mathrm{KL}(\hat{\pi} || \pi^k) = \mathrm{KL}(\hat{\pi} || \hat{\pi}) = 0.$$

Thus we have $\lim_{t \to \infty} \pi^t = \hat{\pi}$ is a Nash equilibrium. This completed the proof of Theorem 1.

# D   PROOF OF THEOREM 2

We show that MWU has linear convergence to the unique Nash equilibrium of a KL-regularized zero-sum game $J(\pi_1, \pi_2, \pi_{\text{ref}})$.

We denote $\mu^\star = \pi_\tau^\star$ the unique Nash equilibrium of the KL regularized game $J\tau(\pi_1, \pi_2, \pi_{\text{ref}})$. We note that $J(\pi_1, \pi_2)$ is 1-smooth. We then can adapt (Abe et al., 2024, Lemma F.1) to our setting.

**Lemma 4** (Adapted from Lemma F.1 in Abe et al. (2024))**.** *If we choose $\eta \in (0, \frac{2\tau}{3\tau^2 + 8}]$, then we have for every $k \geq 1$*

$$\text{KL}(\mu^\star, \mu^{k+1}) \leq (1 - \frac{\eta\tau}{2})\,\text{KL}(\mu^\star, \mu^k).$$

Applying the lemma recursively implies $\text{KL}(\mu^\star || \mu^{k+1}) \leq (1 - \frac{\eta\tau}{2})^k \, \text{KL}(\mu^\star || \pi_{\text{ref}})$ and completes the proof.

# E   COMPUTING THE PROX OPERATOR USING PREFERENCE LEARNING METHODS

We include additional examples showing how existing algorithms designed for RLHF and preference optimization with neural network parameters can be adapted to solve the prox operator $\text{Prox}(\pi, \eta g)$ ($\eta > 0$ is the step size). These algorithms include RL algorithms like PPO and loss-minimization algorithms like DPO, IPO, SPPO, DRO, INPO, each of which may be preferred in certain settings.

**Reinforcement Learning algorithms**   We can use the Proximal Policy Optimization (PPO) algorithm (Schulman et al., 2017) to solve $\text{Prox}(\pi, \eta g)$. Observe that

$$\text{Prox}(\pi, \eta g) = \underset{\pi'}{\text{argmax}}\{\langle \eta g, \pi' \rangle - \text{KL}(\pi' || \pi)\}$$

$$= \underset{\pi'}{\text{argmax}}\, \mathbb{E}_{y \sim \pi'}\big[g[y] - \eta^{-1} \cdot \text{KL}(\pi' || \pi)\big]$$

shares the same form as the objective in (1). Typically, we parameterize $\pi' = \pi_\theta$ with neural network parameters $\theta$ and optimize over $\theta$.

**Loss minimization algorithms**   Let us denote $\hat{\pi}$ the prox operator $\text{Prox}(\pi, \eta g)$, then we have

$$\hat{\pi}[y] = \frac{\pi(y)\exp(\eta g(y))}{Z} \Leftrightarrow \log\frac{\hat{\pi}(y)}{\pi(y)} - \eta g(y) + \log Z = 0,$$

where $Z = \mathbb{E}_{y \sim \pi}[\exp(\eta g(y))]$ is the partition function. We can directly compute the partition function $Z$ and thus $\hat{\pi}$ in small tabular cases. However, the partition function is hard to compute in general large-scale applications. Several works have recently proposed to solve the above equality by optimizing the corresponding $L_2$ loss.

The Self-Play Preference Optimization (SPPO) loss (Wu et al., 2024) assumes $\log Z = \frac{\eta}{2}$ and optimizes

$$\ell_{\text{SPPO}}(\theta) = \left(\log\frac{\pi_\theta(y)}{\pi(y)} - \eta g(y) - \frac{\eta}{2}\right)^2.$$

The Direct Reward Optimization (DRO) loss (Richemond et al., 2024) parameterizes both $\hat{\pi}$ and $\log Z$ with $\theta$ and $V_\phi$ respectively and optimize[6]

$$\ell_{\text{DRO}}(\theta, \phi) = \left(\log\frac{\pi_\theta(y)}{\pi(y)} - \eta g(y) - \eta V_\phi\right)^2.$$

---

[6]we modified some constants in the original DRO loss to make it consistent with our presentation. The modification has no other effects.

The REBEL loss (Gao et al., 2024) uses *differences in rewards* to eliminate the partition function $Z$ and optimize the regression loss

$$\ell_{\text{REBEL}}(\theta) = \left( \eta^{-1} \left( \log \frac{\pi_\theta(y)}{\pi(y)} - \log \frac{\pi_\theta(y')}{\pi(y')} \right) - (g(y) - g(y')) \right)^2.$$

All the above approaches can be used to solve $\text{Prox}(\pi, \eta g)$. However, directly applying them iteratively on $J(\pi_1, \pi_2)$ is equivalent to running MWU, which provably diverges. In contrast, we can apply them in Algorithm 2 and then apply our meta-algorithm COMAL to guarantee convergence to a Nash equilibrium with robust alignment. We present practical implementations of COMAL using the SPPO, DRO, and REBEL loss as subroutines in Appendix J.

**Remark 1.** *The above approaches are versatile and work well for any $g$ that can be evaluated efficiently. In particular, we should consider using them when (1) $g = r$ is a reward function and we can efficiently query $r$; (2) $g = \mathbb{P}(\cdot \mid \mu)$ is the win rate against a reference policy $\mu$, and we can efficiently sample from $\mu$ and have oracle access to $\mathbb{P}$. These two settings are popular and practical in the LLM alignment setting.*

Now we turn attention to the more specific setting where $g$ corresponds to a preference model $\mathbb{P}$ (could be a BT model or a general preference) and that we can collect a win-loss preference data set $\mathcal{D} = \{(y_w, y_l)\}$, which is standard for LLM alignment. Although the abovementioned algorithms apply, they all require estimating $g$ (the win rate) and may be inefficient in practice. In the following, we present algorithms directly working on the sampled dataset $\mathcal{D}$ without further estimation.

**Sampled loss based on the BT preference model**  Assume $g = r$ is the reward of the Bradley-Terry model, and the dataset $\{(y_w, y_l)\}$ consists of win-lose pairs of responses. Then we can solve $\text{Prox}(\pi, \eta g)$ by optimize the DPO loss (Rafailov et al., 2024) defined as

$$\ell_{\text{DPO}}((y_w, y_l); \theta) = -\log \sigma \left( \eta^{-1} \log \frac{\pi_\theta(y_w)}{\pi(y_w)} - \eta^{-1} \log \frac{\pi_\theta(y_l)}{\pi(y_l)} \right).$$

**Sampled loss for general preference**  The DPO loss inspires many other loss functions that work under even weaker assumptions on the preference model. Now, we assume a general preference model $\mathbb{P}$ over $\mathcal{Y}$ (not necessarily the BT model). We assume $g$ is the win-rate against some policy $\mu$ such that $g_\mu(y) = \mathbb{P}[y \succ \mu] := \mathbb{E}_{y' \sim \mu}[\mathbb{P}[y \succ y']]$ (think of $\mu$ as the reference policy $\pi_{\text{ref}}$ or other online policy $\pi_t$). We assume the dataset contains win-lose pairs sampled from $\mu$: $\{y_w, y_l \sim \mu\}$. We denote the preference distribution $\lambda_\mathbb{P}(y, y')$ as a binary distribution:

$$\lambda_\mathbb{P}(y, y') = \begin{cases} (y, y') & \text{with probability } \mathbb{P}[y \succ y'] \\ (y', y) & \text{with probability } 1 - \mathbb{P}[y \succ y'] \end{cases}$$

The (population) IPO loss (Tang et al., 2024; Calandriello et al., 2024) is defined as

$$\ell_{\text{IPO}}(\theta, \mu) := \mathbb{E}_{(y_w, y_l) \sim \mu, (y^+, y^-) \sim \lambda_\mathbb{P}(y_w, y_l)} \left[ \left( \log \frac{\pi_\theta(y^+)}{\pi(y^+)} - \log \frac{\pi_\theta(y^-)}{\pi(y^-)} - \frac{\eta}{2} \right)^2 \right].$$

It has been proved that the minimizer of the $\ell_{\text{IPO}}(\theta, \mu)$ satisfies

$$\pi_\theta(y) \propto \pi(y) \exp\left(-\eta \mathbb{P}[y \succ \mu]\right) \Leftrightarrow \pi_\theta = \text{Prox}(\pi, \eta g_\mu).$$

Thus we can compute the prox operator $\text{Prox}(\pi, \eta g_\mu)$ where $g_\mu = \mathbb{P}(\cdot \succ \mu)$ by minimizing the IPO loss against policy $\mu$.

A variant of the IPO loss applied to the regularized preference setting is the Iterative Nash Policy Optimization (INPO) loss (Zhang et al., 2024). Here, we define $g_\mu^\tau$ the gradient $\nabla_\pi J_\tau(\pi, \mu, \pi_{\text{ref}}) = \mathbb{P}(\cdot \succ \mu) - \tau \log \frac{\mu(\cdot)}{\pi_{\text{ref}}(\cdot)}$ of the regularized objective. The corresponding INPO loss is

$$\ell_{\text{INPO}} := \mathbb{E}_{(y_w, y_l) \sim \mu, (y^+, y^-) \sim \lambda_\mathbb{P}(y_w, y_l)} \left[ \left( \log \frac{\pi_\theta(y^+)}{\pi_\theta(y^-)} - \eta\tau \log \frac{\pi_{\text{ref}}(y^+)}{\pi_{\text{ref}}(y^-)} - (1 - \eta\tau) \log \frac{\mu(y^+)}{\mu(y^-)} - \frac{\eta}{2} \right)^2 \right].$$

Similarly, it has been shown that the INPO loss minimizer corresponds to the prox operator's solution $\text{Prox}(\pi, \eta g_\mu^\tau)$. Thus we can use the INPO in Algorithm 2 directly.

## F  IMPLEMENTATION OF MIRROR-PROX AND OPTIMISTIC MULTIPLICATIVE WEIGHTS UPDATE

We note that there are other algorithms that has provable last-iterate convergence to Nash equilibrium in (unregularized) zero-sum games, including the Mirror-Prox algorithm (Nemirovski, 2004) and Optimistic Multiplicative Weights Update (OMWU) algorithm (Rakhlin and Sridharan, 2013; Syrgkanis et al., 2015; Hsieh et al., 2021). We present practical implementations of these two algorithms in the context of LLM alignment for solving $J(\pi_1, \pi_2)$ (3), where we use preference optimization algorithms to solve the prox operator as shown in Section 3.3 and Appendix E.

We denote the gradient $g(\pi) := \mathbb{P}(\cdot \succ \pi)$.

**Mirror-Prox**   The Mirror-Prox algorithm (Nemirovski, 2004) initialized $\pi^1 = \pi_{\text{sft}}$ and updates in each iteration $t \geq 1$:

$$\pi^{t+\frac{1}{2}} = \text{Prox}(\pi^t, \eta g(\pi^t))$$
$$\pi^{t+1} = \text{Prox}(\pi^t, \eta g(\pi^{t+\frac{1}{2}}))$$

We can implement Mirror-Prox using PPO/DPO/IPO/SPPO/DRO/REBEL to compute the prox operator. Specifically, we could sample from $\pi^t$ and construct a preference dataset $D_t$ and optimize certain regression loss (IPO/DRO/REBEL) to compute $\pi^{t+\frac{1}{2}} = \text{Prox}(\pi^t, \eta g(\pi^t))$. The procedure applies to the second step in each iteration. Thus in such an implementation, we require two sampling and two optimization procedures in each iteration.

**Optimistic Multiplicative Weights Update (OMWU)**   The OMWU algorithm (Rakhlin and Sridharan, 2013) is an optimistic variant of the MWU algorithm. Although MWU diverges in zero-sum games, it has been shown that OMWU has last-iterate convergence to Nash equilibrium (Wei et al., 2021; Hsieh et al., 2021). Initialized with $\pi^1 = \pi^{\frac{1}{2}} = \pi_{\text{sft}}$, OMWU updates in each iteration $t \geq 1$:

$$\pi^{t+\frac{1}{2}} = \text{Prox}(\pi^t, \eta g(\pi^{t-\frac{1}{2}}))$$
$$\pi^{t+1} = \text{Prox}(\pi^t, \eta g(\pi^{t+\frac{1}{2}}))$$

Similarly, we can implement OMWU to solve $J(\pi_1, \pi_2)$ using preference methods to compute the prox operator as shown in Section 3.3. Moreover, OMWU has an equivalent update rule: initialize $\pi^1 = \pi^0 = \pi_{\text{sft}}$

$$\pi^{t+1} = \text{Prox}(\pi^t, 2\eta g(\pi^t) - \eta g(\pi^{t-1})),$$

which requires computing only one prox operator in each iteration.

We leave testing the practical performance of Mirror-Prox and OMWU for large-scale applications, including LLM alignment, as future works.

## G  SETUP FOR SYNTHETIC EXPERIMENTS

Recall that we set $\mathbb{P}[y_b \succ y_a] = \mathbb{P}[y_c \succ y_b] = 0.9$ and $\mathbb{P}[y_a \succ y_c] = 0.8$. This results in the following zero-sum game: we have policies $\Pi = \Delta(\{y_a, y_b, y_c\})$ and objective

$$J(\pi_1, \pi_2) = \pi_1^\top A \pi_2 - 0.5, \text{ where } A = \begin{bmatrix} 0.5 & 0.1 & 0.8 \\ 0.9 & 0.5 & 0.1 \\ 0.2 & 0.9 & 0.5 \end{bmatrix}.$$

The game has a unique Nash equilibrium $[4/11, 3/11, 4/11]$. We set the initial policy to be $\pi^1 = [0.2, 0.5, 0.3]$ for all algorithms. We choose $\eta = 0.3$ for iterative DPO, iterative IPO, and SPPO. We choose $\eta = 0.3$ and $\tau = 0.1$ for INPO and COMAL. For COMAL (Algorithm 4), we set $T = 200$ and $K_t = 25$ so the total number of iterations is $T \cdot K_t = 5000$.

Table 3: Results of the hyperparameter search for DPO and IPO regarding the strength of the KL constraint $\eta^{-1}$. The checkpoints' win rates against the SFT policy are reported.

| $\eta^{-1}$ | IPO | DPO |
|---|---|---|
| 0.02 | 64.34 | 67.32 |
| 0.01 | 69.06 | 69.44 |
| 0.005 | 68.44 | 65.71 |
| 0.002 | 68.94 | 61.49 |
| 0.001 | 58.01 | 53.29 |

## H   HYPERPARAMETER SEARCH FOR REAL-WORLD EXPERIMENTS

Here we outline the results of the hyperparameter search we conducted in §5.1 for identifying the optimal value of $\eta$ for DPO and IPO. Table 3 reports the win rates of different checkpoints trained with different values of $\eta$ against the SFT policy. It shows that DPO achieves the best performance when $\eta^{-1}$ is set to 0.01. On the other hand, IPO achieves a relatively stable and strong performance when $\eta^{-1}$ is set within the range of 0.002-0.01. However, when compared against the best DPO checkpoint, we found that IPO trained with $\eta^{-1} = 0.002$ achieves the highest win rate (51.43%), therefore we chose it as the default value for the rest of the experiments. As discussed in §5.1, the value of the hyper-parameter $\tau$ is determined by setting $\eta\tau$ is set to a fixed ratio, $1/3$, following the setting of Zhang et al. (2024). Due to the high computational cost of the iterative algorithms, we did not perform an extensive search for the optimal value of this ratio. However, our preliminary experiments suggest that the algorithm performance stays relatively stable when the value of this ratio is within the range of 0.1 to 0.5.

## I   ADDITIONAL EVALUATION RESULTS

### I.1   MODEL PERFORMANCE ON STANDARD BENCHMARKS

| Method | BBH | GSM8K | HumanEval | MMLU | Avg |
|---|---|---|---|---|---|
| SFT | 0.3833 | 0.4850 | 0.5730 | 0.5273 | 0.4921 |
| Iter-IPO | 0.4065 | 0.5350 | 0.6187 | 0.5249 | 0.5213 |
| INPO-R3 | 0.4148 | 0.5250 | 0.6130 | 0.5231 | 0.5190 |
| COMAL | 0.4037 | 0.5500 | 0.6100 | 0.5244 | 0.5220 |

Table 4: Performance of various methods across different benchmarks.

Table 4 shows the performance of the checkpoints trained with different algorithms compared in Section 5.2 on standard benchmarks. These include BigBench Hard (BBH) for reasoning (Suzgun et al., 2023), GSM8K for math problem solving (Cobbe et al., 2021), HumanEval for coding (Chen et al., 2021), and MMLU for multi-task language understanding (Hendrycks et al., 2021). The results indicate that the model trained with COMAL achieves similar average performance as the other preference optimization algorithms, while outperforming the baseline SFT model.

### I.2   ADDITIONAL EVALUATION RESULTS WITH GPT-4 AS PREFERENCE ORACLE

As noted in Section 5.1, the preference oracle used for evaluations in Section 5.2, Llama-3-OffsetBias-8B , is the same as the oracle used in model training to ensure the consistency over these two settings. Here, we provide additional evaluation results using GPT-4 (gpt-4-1106-preview) as the preference oracle (i.e., the evaluator), following the default setting of AlpacaEval2.[7]

Table 5 shows the Length-Controlled AlpacaEval Score (Dubois et al., 2024) computed against gpt-4-1106-preview as the baseline systems. Apart from the preference optimization algorithms compared in Section 5.2, two larger LLMs, Llama-2-7B-Chat (Touvron et al., 2023) and Alpaca

---

[7] https://tatsu-lab.github.io/alpaca_eval/

Table 5: Evaluation results on AlpacaEval2 using gpt-4-1106-preview as the evaluator and the baseline system.

| Method/Method | Length-Controlled Score |
|---|---|
| Llama-2-7B-Chat | 5.4 |
| Alpaca 7B | 5.9 |
| SFT | 4.8 |
| Iter-IPO | 4.8 |
| INPO-R3 | 5.6 |
| COMAL | 6.2 |

Table 6: Evaluation results on AlpacaEval2 using gpt-4-1106-preview as the evaluator. The row v.s. column length-controlled AlpacaEval2 scores are reported.

| Row/Col | Iter-IPO | INPO-R3 | COMAL |
|---|---|---|---|
| Iter-IPO | 50.00 | - | 43.55 |
| INPO-R3 | - | 50.00 | 47.08 |
| COMAL | 56.45 | 52.92 | 50.00 |

7B (Taori et al., 2023) are also included for comparison. The results indicate that the checkpoint trained with COMAL not only outperformed other iterative preference optimization algorithms but also the two 7B LLMs. Table 6 presents the direct pairwise comparison results between COMAL and the other iterative preference optimization methods using gpt-4-1106-preview as the evaluator, where COMAL is able to achieve a strictly above 50% win rate. These results further further demonstrate the effectiveness of COMAL over the other algorithms.

## J    MORE PRACTICAL IMPLEMENTATIONS OF COMAL

In this section, we provide more practical implementations of COMAL using the SPPO loss (Wu et al., 2024), the DRO loss (Richemond et al., 2024), and the REBEL loss (Gao et al., 2024). Although these losses are proposed in the unregularized preference setting, we have shown how to extend these losses to compute the prox operator even for KL-regularized preferences in Appendix E. Thus, we can integrate these losses for computing the prox operator in Algorithm 2 for solving the regularized game $J_\tau(\pi_1, \pi_2, \pi_{\text{ref}})$. As a result, we get the practical implementation of COMAL by using different regularized game solvers.

We omit the instruction $x \sim \rho \in \Delta(\mathcal{X})$ for notation simplicity in the following implementations. Generalization to the contextual setting is straightforward.

### J.1    COMAL INTEGRATED WITH SPPO

We present Reg-SPPO (Algorithm 5) for solving a KL regularized game $J_\tau(\pi_1, \pi_2, \pi_{\text{ref}})$, which is the instantiation of Algorithm 2 using the SPPO loss. Then, we give a practical implementation of COMAL integrated with the SPPO loss in Algorithm 6.

### J.2    COMAL INTEGRATED WITH DRO

We present Reg-DRO (Algorithm 7) for solving a KL regularized game $J_\tau(\pi_1, \pi_2, \pi_{\text{ref}})$, which is the instantiation of Algorithm 2 using the DRO loss. Then, we give a practical implementation of COMAL integrated with the DRO loss in Algorithm 8.

---

**Algorithm 5:** Reg-SPPO: Extension of SPPO (Wu et al., 2024) for solving KL-regularized games

---

**Input:** Reference policy $\pi_{\text{ref}}$, regularization $\tau > 0$, step size $\eta > 0$, number of rounds $K \geq 1$, preference oracle $\mathbb{P}$.

**Output:** Approximate regularized Nash equilibrium policy $\mu_K$

Initialize $\mu^1 \leftarrow \pi_{\text{ref}}$

**for** $k = 1, 2, \ldots, K - 1$ **do**

    Generate responses $\{y^{(i)} \sim \mu^k\}_{i=1}^n$

    Query preference oracle $\mathbb{P}$ to annotate the win-rate $\mathbb{P}[y^{(i)} \succ y^{(j)}], \forall i, j \in [n]$

    Form dataset $\mathcal{D}_t = \{(y^{(i)}, \widehat{P}[y^{(i)} \succ \mu^k])\}_{i \in [n]}$

    Compute $\mu^{k+1} = \mu_{\theta^{k+1}}$ where

$$\theta^{k+1} = \operatorname*{argmin}_{\theta} \ell_{\text{SPPO}}(\theta) := \mathbb{E}_{(y, \widehat{P}[y \succ \mu^k]) \sim \mathcal{D}_t}\left[\left(\log \frac{\mu_\theta(y)}{\mu^k(y)} - \eta\left(\widehat{P}[y \succ \mu^k] - \tau \log \frac{\mu^k(y)}{\pi_{\text{ref}}(y)} - \frac{1}{2}\right)\right)^2\right]$$

**return** $\mu^K$

---

---

**Algorithm 6:** Practical Implementation of COMAL integrated with Reg-SPPO (Algorithm 5)

---

**Input:** Initial policy $\pi_{\text{sft}}$, regularization $\{\tau_t > 0\}$, step size $\{\eta_t > 0\}$, number of iterations $T \geq 1$, number of inner optimization steps $\{K_t \geq 1\}$, preference oracle $\mathbb{P}$.

**Output:** Optimized policy $\pi^T$

Initialize $\pi^1, \pi_{\text{ref}} \leftarrow \pi_{\text{sft}}$

**for** $t = 1, 2, \ldots, T - 1$ **do**

    $\pi^{t+1} \leftarrow \text{Reg-SPPO}(\pi_{\text{ref}}, \tau_t, \eta_t, K_t, \mathbb{P})$ defined in Algorithm 5

    $\pi_{\text{ref}} \leftarrow \pi^{t+1}$

**return** $\pi^T$

---

---

**Algorithm 7:** Reg-DRO: Extension of DRO (Richemond et al., 2024) for solving KL-regularized games

---

**Input:** Reference policy $\pi_{\text{ref}}$, regularization $\tau > 0$, step size $\eta > 0$, number of rounds $K \geq 1$, preference oracle $\mathbb{P}$.

**Output:** Approximate regularized Nash equilibrium policy $\mu_K$

Initialize $\mu^1 \leftarrow \pi_{\text{ref}}$

**for** $k = 1, 2, \ldots, K - 1$ **do**

    Generate responses $\{y^{(i)} \sim \mu^k\}_{i=1}^n$

    Query preference oracle $\mathbb{P}$ to annotate the win-rate $\mathbb{P}[y^{(i)} \succ y^{(j)}], \forall i, j \in [n]$

    Form dataset $\mathcal{D}_t = \{(y^{(i)}, \widehat{P}[y^{(i)} \succ \mu^k])\}_{i \in [n]}$

    Compute $\mu^{k+1} = \mu_{\theta^{k+1}}$ where

$$\theta^{k+1} = \operatorname*{argmin}_{\theta} \min_{\phi} \ell_{\text{DRO}}(\theta) := \mathbb{E}_{(y, \widehat{P}[y \succ \mu^k]) \sim \mathcal{D}_t}\left[\left(\log \frac{\mu_\theta(y)}{\mu^k(y)} - \eta\left(\widehat{P}[y \succ \mu^k] - \tau \log \frac{\mu^k(y)}{\pi_{\text{ref}}(y)}\right) - \eta V_\phi\right)^2\right]$$

**return** $\mu^K$

---

---

**Algorithm 8:** Practical Implementation of COMAL integrated with Reg-DRO (Algorithm 7)

---

**Input:** Initial policy $\pi_{\text{sft}}$, regularization $\{\tau_t > 0\}$, step size $\{\eta_t > 0\}$, number of iterations $T \geq 1$, number of inner optimization steps $\{K_t \geq 1\}$, preference oracle $\mathbb{P}$.

**Output:** Optimized policy $\pi^T$

Initialize $\pi^1, \pi_{\text{ref}} \leftarrow \pi_{\text{sft}}$

**for** $t = 1, 2, \ldots, T - 1$ **do**

    $\pi^{t+1} \leftarrow \text{Reg-DRO}(\pi_{\text{ref}}, \tau_t, \eta_t, K_t, \mathbb{P})$ defined in Algorithm 5

    $\pi_{\text{ref}} \leftarrow \pi^{t+1}$

**return** $\pi^T$

---

## J.3 COMAL INTEGRATED WITH REBEL

We present Reg-REBEL (Algorithm 9) for solving a KL regularized game $J_\tau(\pi_1, \pi_2, \pi_{\text{ref}})$, which is the instantiation of Algorithm 2 using the REBEL loss. Then, we give a practical implementation of COMAL (Algorithm 1) integrated with the REBEL loss in Algorithm 10.

---

**Algorithm 9:** Reg-REBEL: Extension of REBEL (Gao et al., 2024) for solving KL-regularized games

---

**Input:** Reference policy $\pi_{\text{ref}}$, regularization $\tau > 0$, step size $\eta > 0$, number of rounds $K \geq 1$, preference oracle $\mathbb{P}$.

**Output:** Approximate regularized Nash equilibrium policy $\mu_K$

Initialize $\mu^1 \leftarrow \pi_{\text{ref}}$

**for** $k = 1, 2, \ldots, K - 1$ **do**

    Generate responses $\{y^{(i)} \sim \mu^k\}_{i=1}^n$

    Query preference oracle $\mathbb{P}$ to annotate the win-rate $\mathbb{P}[y^{(i)} \succ y^{(j)}], \forall i, j \in [n]$

    Form dataset $\mathcal{D}_t = \{(y^{(i)}, y^{(j)}, \widehat{P}[y^{(i)} \succ \mu^k], \widehat{P}[y^{(j)} \succ \mu^k])\}_{i,j \in [n]}$

    Compute $\mu^{k+1} = \mu_{\theta^{k+1}}$ where

$$\theta^{k+1} = \underset{\theta}{\arg\min} \, \ell_{\text{REBEL}}(\theta)$$

$$\ell_{\text{REBEL}}(\theta) := \mathbb{E}_{(y,y') \sim \mathcal{D}_t}\left[\left(\left(\eta^{-1}\left(\log\frac{\mu_\theta(y)}{\mu^k(y)} - \log\frac{\mu_\theta(y')}{\mu^k(y')}\right) - \left(\widehat{P}[y \succ \mu^k] - \tau\log\frac{\mu^k(y)}{\pi_{\text{ref}}(y)} - \widehat{P}[y' \succ \mu^k] + \tau\log\frac{\mu^k(y')}{\pi_{\text{ref}}(y')}\right)\right)^2\right]$$

**return** $\mu^K$

---

---

**Algorithm 10:** Practical Implementation of COMAL integrated with Reg-REBEL (Algorithm 9)

---

**Input:** Initial policy $\pi_{\text{sft}}$, regularization $\{\tau_t > 0\}$, step size $\{\eta_t > 0\}$, number of iterations $T \geq 1$, number of inner optimization steps $\{K_t \geq 1\}$, preference oracle $\mathbb{P}$.

**Output:** Optimized policy $\pi^T$

Initialize $\pi^1, \pi_{\text{ref}} \leftarrow \pi_{\text{sft}}$

**for** $t = 1, 2, \ldots, T - 1$ **do**

    $\pi^{t+1} \leftarrow$ Reg-REBEL$(\pi_{\text{ref}}, \tau_t, \eta_t, K_t, \mathbb{P})$ defined in Algorithm 9

    $\pi_{\text{ref}} \leftarrow \pi^{t+1}$

**return** $\pi^T$

---

