# OpenReview forum: "COMAL: A Convergent Meta-Algorithm for Aligning LLMs with General Preferences"
_ICLR.cc/2025/Conference — Submitted to ICLR 2025_

### Official Review · Reviewer_tugW · 2024-10-21

**Soundness:** 2
**Presentation:** 2
**Contribution:** 1
**Rating:** 3
**Confidence:** 4

**Summary:**

This paper proposes a new algorithm, Convergent Meta Alignment Algorithm (COMAL), aimed at aligning language models with general preferences. The authors frame the alignment promblem as a two-player zero-sum game and demonstrate that COMAL can converge to a "exact" Nash equilibrium strategy using prox operaters, ensuring a minimum 50% win rate under any competing strategy, which they refer to as "robust alignment". The paper provides theoretical guarantees for the algorithm’s convergence and demonstrates its advantages over existing methods through synthetic and real-world LLM fine-tuning experiments.

**Strengths:**

The main strengths of this paper are:

1. introduces a new concept called robust alignment and provides a theoretical gurantee that COMAL can achieve robust alignment, whereas existing methods cannot.

2. COMAL can be easily integrated into existing algorithms, such as INPO, to achieve robust alignment.

**Weaknesses:**

1. The paper introduces a novel concept called "robust alignment". However, the motivation behind this concept is unclear and lacks a rigorous mathematical definition. The authors state that robust alignment aims to ensure a minimum 50% win rate under any competing strategy, yet the connection between this concept and alignment is not well-established. Furthermore, it is not evident how the "robust" aspect is manifested in this context. The significance of achieving robust alignment in the broader context of alignment research remains ambiguous. From my perspective, this concept appears to be more of a repackaging of existing game theory results [1] rather than a substantive contribution.

2. The main idea of this paper stems from [1] and [2], which established the result of last-iterate convergence to the exact Nash equilibrium in monotone games. However, this paper lacks proper citation of these two works. Even in the section on last-iterate convergence in the related work, [1] is not cited, and no comparison or discussion is provided. This oversight fails to acknowledge the foundational contributions of these prior works.

3. The authors’ emphasis on the prox operator as the primary mechanism for achieving last-iterate convergence is confusing and misleading. It appears that the authors may not fully understand the theoretical foundations of the results in [1] and [2]. In fact, last-iterate convergence is achieved through the introduction of an additional convex regularizer in the payoff, rather than by the prox operator itself. While Mirror Descent (MD) is a proximal algorithm, MD alone does not ensure last-iterate convergence.  This means than COMAL actually does not apply to Iterative DPO, Iterative IPO, SPPO.

4. This paper builds on the perspective introduced by SPIN [3], which views DPO as the first iteration of MD. However, according to [1], for last-iterate convergence to the exact NE, it is necessary to first converge to the stationary point of the regularized game. Thus, as highlighted in [1], it is crucial to run the iterations for a sufficiently large number of iterations before updating the slingshot strategy. However, COMAL replaces the reference strategy after just a single iteration, which I believe is insufficient for convergence to the NE of the regularized game. Even from a practical standpoint, this falls significantly short. Given that this is the only modification COMAL makes to existing algorithms, I consider it a significant flaw in the framework.

5. The experiments in this paper are extremely limited. The first experiment is overly simplistic and does not address real-world issues. The authors merely tested MD on a simple matrix game, but the results are already well-known and are far different from alignment scenarios, providing little insight into how COMAL addresses alignment challenges. The second experiment has two major issues: first, the model used, Qwen2-1.5B, is too small to be meaningful—at least an 7B parameter model should have been used to demonstrate the effectiveness. Second, while the authors used the AlpacaEval dataset, the evaluation relied on Llama-3-OffsetBias-8B, which fails to convincingly show that COMAL can meaningfully improve LLM performance. To truly demonstrate COMAL’s effectiveness, results showing consistent improvements across iterations are necessary ( since the authors claims last-iterate convergence).  Besides, the model's performance should be evaluated in at least two widely used benchmarks to demonstrate its effectiveness. Without such evidence, the results presented are insufficient to support the claims.

In summary, considering that the core idea of COMAL primarily stems from [1] and [2], but the paper lacks proper citations and comparisons, while also showing a lack of understanding of the insights from [1] and [2] and how these insights apply to alignment. Furthermore, COMAL only introduces minimal modifications to existing methods, and the experiments provided are unconvincing. The overall contribution of the paper appears limited both in terms of theoretical novelty and empirical validation.

[1] Abe, K., Ariu, K., Sakamoto, M., & Iwasaki, A. Adaptively Perturbed Mirror Descent for Learning in Games. In Forty-first International Conference on Machine Learning.

[2] Perolat, J., Munos, R., Lespiau, J. B., Omidshafiei, S., Rowland, M., Ortega, P., ... & Tuyls, K. (2021, July). From poincaré recurrence to convergence in imperfect information games: Finding equilibrium via regularization. In International Conference on Machine Learning (pp. 8525-8535). PMLR.

[3] Chen, Z., Deng, Y., Yuan, H., Ji, K., & Gu, Q. (2024). Self-play fine-tuning converts weak language models to strong language models. arXiv preprint arXiv:2401.01335.

**Questions:**

How is COMAL’s performance on the standard AlpacaEval benchmark? Since the responses have already been generated in the second part of the experiment, it should only require following the standard AlpacaEval pipeline and using GPT-4 for evaluation to assess the performance.

---

> ### Author Response · Authors · 2024-11-21
> **Response to Reviewer tugW [1/4]**
>
> Thank you for your review.
>
> > “The paper introduces a novel concept called "robust alignment". However, the motivation behind this concept is unclear and lacks a rigorous mathematical definition. The authors state that robust alignment aims to ensure a minimum 50% win rate under any competing strategy, yet the connection between this concept and alignment is not well-established. Furthermore, it is not evident how the "robust" aspect is manifested in this context. The significance of achieving robust alignment in the broader context of alignment research remains ambiguous. From my perspective, this concept appears to be more of a repackaging of existing game theory results [1] rather than a substantive contribution.”
>
> We define a policy as having ‘robust alignment’ if it achieves at least a 50% win rate against any other policy (Lines 050-052). We clarify that we never claim this concept as a substantive contribution. This concept is not new or novel; it is a natural property of the Nash equilibrium policy [2]. The term ‘robust’ reflects the fact that no other policy can outperform this policy more than half of the time. This property is highly desirable for aligning LLMs with general preferences and is a focus of many recent studies [2, 3].
>
> References
>
> [1] Abe, K., Ariu, K., Sakamoto, M., & Iwasaki, A. Adaptively Perturbed Mirror Descent for Learning in Games. In Forty-first International Conference on Machine Learning.
>
> [2] Munos, Remi, Michal Valko, Daniele Calandriello, Mohammad Gheshlaghi Azar, Mark Rowland, Zhaohan Daniel Guo, Yunhao Tang et al. "Nash Learning from Human Feedback." In Forty-first International Conference on Machine Learning.
>
> [3] Swamy, Gokul, Christoph Dann, Rahul Kidambi, Steven Wu, and Alekh Agarwal. "A Minimaximalist Approach to Reinforcement Learning from Human Feedback." In Forty-first International Conference on Machine Learning.
>
> ---
>
> > “The main idea of this paper stems from [1] and [2], which established the result of last-iterate convergence to the exact Nash equilibrium in monotone games. However, this paper lacks proper citation of these two works. Even in the section on last-iterate convergence in the related work, [1] is not cited, and no comparison or discussion is provided. This oversight fails to acknowledge the foundational contributions of these prior works.”
>
> We respectfully disagree with the reviewer’s comment regarding the lack of proper citation and discussion of [1] and [2]. Both works are already explicitly cited and discussed in the **main body** of our paper. On page 4, lines 180–184, we state:
> “COMAL (Algorithm 1) is an online iterative algorithm inspired by the classic conceptual prox method (Nemirovski, 2004), first introduced in the optimization theory community. This method has recently been applied to finding a Nash equilibrium in zero-sum games (Perolat et al., 2021; Abe et al., 2024) and has had notable success in training advanced game AI models (Perolat et al., 2022).” We will also include it in the related works section in the appendix.
>
> We would like to remark that the conceptual prox algorithm is proposed for solving variational inequality problems, which cover two-player zero-sum games as a special case. The algorithms proposed in both [1] and [2] are inspired by and are variants of the conceptual prox algorithms instantiated with different regularizers. We have acknowledged the contributions of [1] and [2], highlighting their application of the conceptual prox method to solving zero-sum games. Our work is along the same line, but we extend the conceptual prox algorithm to the domain of LLM alignment, which introduces challenges in computing the prox operator in large-scale applications. Moreover, we conducted LLM-based experiments and showed their effectiveness, which had not been done before. That said, we are happy to incorporate further details if there are specific aspects where additional discussion or comparison would be helpful.
>
> References
>
> [1] Abe, K., Ariu, K., Sakamoto, M., & Iwasaki, A. Adaptively Perturbed Mirror Descent for Learning in Games. In Forty-first International Conference on Machine Learning.
>
> [2] Perolat, J., Munos, R., Lespiau, J. B., Omidshafiei, S., Rowland, M., Ortega, P., ... & Tuyls, K. (2021, July). From poincaré recurrence to convergence in imperfect information games: Finding equilibrium via regularization. In International Conference on Machine Learning (pp. 8525-8535). PMLR.
>
> [Nemirovski, 2004] Nemirovski, Arkadi. "Prox-method with rate of convergence O (1/t) for variational inequalities with Lipschitz continuous monotone operators and smooth convex-concave saddle point problems." SIAM Journal on Optimization 15.1 (2004): 229-251.

---

> ### Author Response · Authors · 2024-11-21
> **Response to Reviewer tugW [2/4]**
>
> > “The authors’ emphasis on the prox operator as the primary mechanism for achieving last-iterate convergence is confusing and misleading. It appears that the authors may not fully understand the theoretical foundations of the results in [1] and [2]. In fact, last-iterate convergence is achieved through the introduction of an additional convex regularizer in the payoff, rather than by the prox operator itself. While Mirror Descent (MD) is a proximal algorithm, MD alone does not ensure last-iterate convergence. This means than COMAL actually does not apply to Iterative DPO, Iterative IPO, SPPO.”
>
> We must respectfully but firmly disagree with the reviewer's assessment that we "may not fully understand the theoretical foundations" of [1] and [2]. This statement is incorrect and appears to stem from a misreading of our work. We would like to assure the reviewer that we have a solid understanding of the conceptual prox algorithm presented in [Nemirovski 2004], as well as the more recent results [1, 2].
>
> We’d like to clarify that our main contribution is introducing the conceptual prox algorithm to LLM alignment and developing the first last-iterate convergent algorithm, COMAL. Several recent algorithms, including RLHF, DPO, IPO, SPPO, and INPO, use the prox operator for LLM alignment, although not explicitly stated sometimes. Unfortunately, these algorithms either diverge or converge only in a modified game. **To sum up, using the prox operator does not guarantee last-iterate convergence.** COMAL uses the prox-operator in a more sophisticated way inspired by the conceptual-prox algorithm and guarantees last-iterate convergence.
>
> Let us clarify our algorithm. As detailed in Algorithm 1 of our paper, each iteration involves the following two steps: (1) finding the regularized Nash equilibrium of the KL-regularized game with respect to the reference policy using Mirror Descent (MD), as outlined in Algorithm 2, and (2) updating the reference policy to the Nash equilibrium of this regularized game. MD, as we demonstrate, exhibits exponentially fast convergence in this regularized setting (Theorem 2), allowing for efficient computation of the regularized Nash equilibrium. By iteratively updating the reference policy, our approach establishes equivalence to the conceptual prox algorithm, as per [Nemirovski 2004], ultimately ensuring convergence to the exact Nash equilibrium (Theorem 1).
>
> We note that to run Algorithm 1, we only need to implement the prox operator step in MD (Algorithm 2). Section 3.2 and Appendix E show how existing methods, such as DPO loss, IPO loss,  SPPO loss, DRO loss, and REBEL loss, can be **adapted** to compute the prox operator. We note that the DPO loss works only for BT preferences, so it may not be applied to general preferences. However, IPO, SPPO, DRO, and REBEL losses could be adapted (not directly because they are proposed for unregularized preferences) to general preferences with KL regularization:  we just need to set the $g$ in the losses (defined in Appendix E Line 1060-1074) with the regularized gradient $g^k_\tau$ (Line 207), which can be estimated from samples. So we can adapt these losses as subroutines for COMAL. As a concrete example, we provide a practical implementation of COMAL (Algorithm 4) using the INPO loss,which is the IPO loss adapted for the KL regularized preference. We will add further discussions about this in the revised version.
>
>
> References
>
> [1] Abe, K., Ariu, K., Sakamoto, M., & Iwasaki, A. Adaptively Perturbed Mirror Descent for Learning in Games. In Forty-first International Conference on Machine Learning.
>
> [2] Perolat, J., Munos, R., Lespiau, J. B., Omidshafiei, S., Rowland, M., Ortega, P., ... & Tuyls, K. (2021, July). From poincaré recurrence to convergence in imperfect information games: Finding equilibrium via regularization. In International Conference on Machine Learning (pp. 8525-8535). PMLR.
>
> [Nemirovski, 2004] Nemirovski, Arkadi. "Prox-method with rate of convergence O (1/t) for variational inequalities with Lipschitz continuous monotone operators and smooth convex-concave saddle point problems." SIAM Journal on Optimization 15.1 (2004): 229-251.

---

> ### Author Response · Authors · 2024-11-21
> **Response to Reviewer tugW [3/4]**
>
> > “This paper builds on the perspective introduced by SPIN [1], which views DPO as the first iteration of MD. However, according to [2], for last-iterate convergence to the exact NE, it is necessary to first converge to the stationary point of the regularized game. Thus, as highlighted in [2], it is crucial to run the iterations for a sufficiently large number of iterations before updating the slingshot strategy. However, COMAL replaces the reference strategy after just a single iteration, which I believe is insufficient for convergence to the NE of the regularized game. Even from a practical standpoint, this falls significantly short. Given that this is the only modification COMAL makes to existing algorithms, I consider it a significant flaw in the framework.”
>
> The claim that “COMAL replaces the reference strategy after just a single iteration” is incorrect and a severe misunderstanding of our algorithm.
>
> As described in Algorithm 1 (Line 196), we perform several iterations of Algorithm 2 to ensure convergence to the regularized Nash equilibrium (which corresponds to the stationary point mentioned in the review) before updating the reference policy. This iterative process is also explicitly detailed in our practical implementation (Algorithm 4), where we run  $K_t$ steps of INPO before updating the reference policy. Moreover, our work provides a provable last-iterate convergence guarantee established in Theorems 1 and 2. Additionally, in our LLM-based experiments, we conducted 12 iterations before updating the reference policy and showed strong performance of COMAL. To sum up, COMAL (Algorithm 1) is a sound approach for LLM alignment with both provable last-iterate convergence and strong practical performance.
>
> References
>
> [1] Chen, Z., Deng, Y., Yuan, H., Ji, K., & Gu, Q. (2024). Self-play fine-tuning converts weak language models to strong language models. arXiv preprint arXiv:2401.01335.
>
> [2] Abe, K., Ariu, K., Sakamoto, M., & Iwasaki, A. Adaptively Perturbed Mirror Descent for Learning in Games. In Forty-first International Conference on Machine Learning.

---

> ### Author Response · Authors · 2024-11-21
> **Response to Reviewer tugW [4/4]**
>
> > “The experiments in this paper are extremely limited. The first experiment is overly simplistic and does not address real-world issues. The authors merely tested MD on a simple matrix game, but the results are already well-known and are far different from alignment scenarios, providing little insight into how COMAL addresses alignment challenges. The second experiment has two major issues: first, the model used, Qwen2-1.5B, is too small to be meaningful—at least an 7B parameter model should have been used to demonstrate the effectiveness. Second, while the authors used the AlpacaEval dataset, the evaluation relied on Llama-3-OffsetBias-8B, which fails to convincingly show that COMAL can meaningfully improve LLM performance. To truly demonstrate COMAL’s effectiveness, results showing consistent improvements across iterations are necessary (since the authors claims last-iterate convergence). Besides, the model's performance should be evaluated in at least two widely used benchmarks to demonstrate its effectiveness. Without such evidence, the results presented are insufficient to support the claims.”
>
> 1. Our synthetic experiment provides a transparent and controlled way to compare different algorithms.
>
> 2. **Concern regarding the model size**: we would appreciate references to related studies that support the claim that “the model used, Qwen2-1.5B, is too small to be meaningful—at least a 7B parameter model should have been used to demonstrate the effectiveness.” We note that Qwen2-1.5B [1] can outperform Llama-2-7B [2] on important benchmarks such as MMLU, MATH, GSM8K, HumanEval. Therefore, without concrete evidence, we find it hard to agree with the previous statement. As a reference, Phi-1.5 [3] is a 1.3B model that shows strong capabilites in instruction-following. We would also like to emphasize that, as we noted in Lines 461 - 467, it takes around 700 GPU hours in total to finish the training of 18 iterations. In contrast, previous work such as INPO [4] usually just performed around 3-round training. Consequently, the high computational cost makes it difficult for us to experiment with larger models. That said, we are indeed interested in extending experiments with larger models, and plan to include that in the revised version.
>
> 3. **Evaluations on more benchmarks**: Thank you for the suggestions. We have conducted an additional evaluation with GPT-4 as the evaluator on AlpacaEval (detailed in the next response), which further demonstrates the effectiveness of our algorithm. We note that since AlpacaEval contains a wide range of instruction types, the evaluations conducted on AlpacaEval are likely to be generalizable.
>
> 4. **Evaluation across iterations**: First, our theory guarantees hold for the KL divergence between the iterate and the Nash equilibrium, which we can not measure for LLM experiments because it is out of reach to compute the exact Nash equilibrium. Moreover, a consistent decrease in the distance to the Nash equilibrium does not imply that the win rate against a specific policy or the performance on the benchmarks is monotonically increasing. That said, an important property of Nash equilibrium is at least 50\% against any other policies. Our experiments show that COMAL achieves a strictly greater than 50\% win rate against all baseline algorithms, showing its superiority over other iterative algorithms in approximating the Nash equilibrium.
>
>
> References
>
> [1] Yang, An, et al. "Qwen2 technical report." arXiv preprint arXiv:2407.10671 (2024).
>
> [2] Touvron, Hugo, et al. "Llama 2: Open foundation and fine-tuned chat models." arXiv preprint arXiv:2307.09288 (2023).
>
> [3] Li, Yuanzhi, et al. "Textbooks are all you need ii: phi-1.5 technical report." arXiv preprint arXiv:2309.05463 (2023).
>
> [4] Zhang, Yuheng, et al. "Iterative Nash Policy Optimization: Aligning LLMs with General Preferences via No-Regret Learning." arXiv preprint arXiv:2407.00617 (2024).
>
> ---
>
> > “How is COMAL’s performance on the standard AlpacaEval benchmark? Since the responses have already been generated in the second part of the experiment, it should only require following the standard AlpacaEval pipeline and using GPT-4 for evaluation to assess the performance.”
>
> Thank you for the question. Following your advice, we have used GPT-4 to compare different training algorithms on AlpacaEval. The win rates are as follows: Iter-IPO v.s. COMAL: 43.55 v.s. 56.45; INPO (Round 3) v.s COMAL: 47.08 v.s. 52.92. The length-controlled win rates [1] are as follows: Iter-IPO v.s. COMAL: 43.10 v.s. 56.90; INPO (Round 3) v.s COMAL: 46.81 v.s. 53.19. These results further demonstrate the effectiveness of our proposed algorithm. We will include these results in the revised version.
>
> Reference
>
> [1] Dubois, Yann, et al. "Length-controlled alpacaeval: A simple way to debias automatic evaluators." COLM 2024.

---

> ### Comment · Reviewer_tugW · 2024-11-22
>
> Thanks for your reply. However, most of my concerns are not well addressed. And the overall presentation is confusing and misleading.
> 1. As the authors pointed out,  a policy that achieves ‘robust alignment’ is essentially the Nash equilibrium policy as defined in [1]. This Nash equlibrium corresponds to the solution concept known as the Minmax Winner in social choice theory [1]. I do not see the necessity of introducing ‘robust alignment,’ as it does not provide any new insights to the current alignment field and merely rebrands the already well-established Minmax Winner solution concept. This repackaging only confuses the readers.
> 2. As the authors stated, the conceptual prox algorithm is proposed for solving variational inequality problems. And this method serves as a general framework for addressing various issues. Introducing this method in the context of alignment does not offer any new insights specific to the alignment problem, nor does it inherently imply the desired property of convergence to the exact Nash equilibrium. Presenting this method only confuses and misleads readers about the actual mechanisms necessary for achieving exact NE convergence—mechanisms that have already been developed and rigorously detailed in [2][3]. Additionally, the proof of COMAL does not introduce any new techniques beyond those already established in [2][3], further emphasizing the lack of novel contributions in this regard.
> 3. As noted in the second point, the core idea of this paper is largely derived from recent results in [2][3], particularly the results in [3]. However, in the ‘Last-Iterate Convergence on Games’ section of the Related Work, the authors fail to mention or compare their work with these two most closely related studies. This omission is particularly surprising given that, as the authors highlighted in the reply, in lines 180–184, the authors acknowledge the importance of these works. And Lemma 4 in the appendix is directly adapted from Lemma F.1 in [3]. Such a lack of engagement with directly relevant prior research undermines the contextual rigor and completeness of the paper.
> 4. While Algorithm 1 (Line 196) is conceptually correct, the iterative process described in the practical implementation of Algorithm 4 is inconsistent with Algorithm 1, and from my perspective, the differences are significant. Algorithm 1 suggests that to converge to the exact Nash equilibrium (NE), it is necessary first to converge to the NE of the KL-regularized game and then update the reference policy to this NE. However, in line 345 of Algorithm 4, the opponent is fixed, and INPO only learns a single best response to the opponent. This learned policy represents a best response rather than the NE of the KL-regularized game. Moreover, line 345 in Algorithm 4 corresponds to a single iteration of mirror descent (i.e., $\pi_t \to \pi_{t+1}$, not the gradient update step in the authors’ reply). Whether this single iteration can produce a meaningful response is highly questionable, let alone achieve convergence to the NE of the KL-regularized game. And the approximation error is not studied. This significant inconsistency between Algorithm 1 and Algorithm 4 is misleading, making the implementation in Algorithm 4 lack solid theoretical foundations.
> 5. The IPO, SPPO, DRO, and REBEL losses cannot be adapted to this framework in their original forms, as they use MD to solve the original game rather than the KL-regularized game. Consequently, the theoretical guarantees presented in the paper are applicable only to INPO, not to these other loss functions. This limitation should be explicitly acknowledged to avoid overstating the generality of the framework.
> 6. Adhering to the standard AlpacaEval pipeline [4] is essential. Specifically, evaluations should be conducted against GPT-4-turbo as baseline, rather than relying on pairwise comparisons against the trained models mentioned in the reply. This approach would provide readers with a more comprehensive understanding of COMAL’s performance on the standard benchmark and enable direct comparisons with other opensource strong baselines featured on the open leaderboard [5].
>
> References
>
> [1] Swamy, Gokul, et al. "A minimaximalist approach to reinforcement learning from human feedback." arXiv preprint arXiv:2401.04056 (2024).
>
> [2] Perolat, Julien, et al. "From poincaré recurrence to convergence in imperfect information games: Finding equilibrium via regularization." International Conference on Machine Learning. PMLR, 2021.
>
> [3] Abe, Kenshi, et al. "Adaptively Perturbed Mirror Descent for Learning in Games." Forty-first International Conference on Machine Learning.
>
> [4] Dubois, Yann, et al. "Length-controlled alpacaeval: A simple way to debias automatic evaluators." arXiv preprint arXiv:2404.04475 (2024).
>
> [5] https://tatsu-lab.github.io/alpaca_eval/

---

> > ### Author Response · Authors · 2024-11-24
> > **Response to Reviewer tugW’s comment [2/6]**
> >
> > **Proposition 2: IPO, SPPO, DRO, and REBEL losses could be adapted (_not directly_ because they are proposed for unregularized preferences) to general preferences with KL regularization. So we can adapt these losses as subroutines for COMAL.**
> >
> > This proposition is a direct quotation from two sentences in our previous response.
> >
> > The reviewer’s comment regarding Proposition 2:
> > > “5. The IPO, SPPO, DRO, and REBEL losses cannot be adapted to this framework in their original forms, as they use MD to solve the original game rather than the KL-regularized game. Consequently, the theoretical guarantees presented in the paper are applicable only to INPO, not to these other loss functions. This limitation should be explicitly acknowledged to avoid overstating the generality of the framework.”
> >
> > Our response:
> > 1. We agree with the assertion that “the IPO, SPPO, DRO, and REBEL losses cannot be adapted to this framework in their original forms,” as previously clarified in our response.
> >
> > 2. We suspect a confusion/miscommunication of the word “adapt” used in the paper and the first response: the reviewer might think “adapt” means applying the losses without any changes, while we use “adapt” to mean these losses could be extended/generalized to KL-regularized preferences with slight modifications. If this is indeed the case, we are happy to rephrase our statement as follows: *“IPO, SPPO, DRO, and REBEL losses could be extended to general preferences with KL regularization.”* We would like to reiterate that, both in the paper and in our response, we never claim that these losses can be directly applied without any modifications.
> >
> > 3. Next, we illustrate how the generalizations to KL-regularized preferences work. In the paper and the first round of responses, we have shown that adaptation of the (population) IPO loss to KL-regularized preferences is precisely the INPO loss, which has been observed and proved in [1]. We have also demonstrated how to adapt SPPO, DRO, and REBEL losses from unregularized preference to KL regularized preference in our previous response: *“We just need to set the $g$ in the losses (defined in Appendix E Line 1060-1074) to the regularized gradient $g^k_\tau$ (Line 207), which can be estimated from samples.”* Minimizing these losses implements the prox operator for regularized preference as required in Lines 207-208. Moreover, as stated in the first response, we will include more details in the revised version. Here, we confirm that we will include detailed information, such as the practical implementation of COMAL using the generalized SPPO loss. This update will occur when we revise our submission to incorporate the author-reviewer discussion by the end of the discussion period.
> >
> > Reference
> >
> > [1] Zhang, Yuheng, et al. "Iterative Nash Policy Optimization: Aligning LLMs with General Preferences via No-Regret Learning." arXiv preprint arXiv:2407.00617 (2024).

---

> > ### Author Response · Authors · 2024-11-24
> > **Response to Reviewer tugW’s comment [3/6]**
> >
> > **Response to other comments:**
> >
> > > “1. As the authors pointed out, a policy that achieves ‘robust alignment’ is essentially the Nash equilibrium policy as defined in [1]. This Nash equlibrium corresponds to the solution concept known as the Minmax Winner in social choice theory [1]. I do not see the necessity of introducing ‘robust alignment,’ as it does not provide any new insights to the current alignment field and merely rebrands the already well-established Minmax Winner solution concept. This repackaging only confuses the readers.”
> >
> > 1. As we noted in our previous response, we do not regard the introduction of the term “robust alignment” as a major contribution of this work; we coined it simply for ease of discussion. Therefore, the reviewer’s critique regarding repackaging is minor, as our intent was not to repackage but to facilitate understanding. “Robust alignment” is clearly defined (Lines 050 - 052) as a policy’s property that guarantees at least a 50% win rate against any other policy. We use this term specifically in the context of preference optimization/alignment fine-tuning. Moreover, the concept of robust alignment clearly distinguishes between the Nash equilibrium (NE) of the unregularized game and the NE of the KL-regularized game, since only the former NE guarantees a 50% win rate against any other policies. While we believe the definition of robust alignment is clear to our readers, we are open to considering other terms and suggestions to enhance clarity.
> >
> > 2. The alternative term “Minimax Winner” suggested by the reviewer seems to be defined in the reference they provided [1]: “we define the Minimax Winner (MW, Kreweras (1965); Simpson (1969); Kramer (1973); Fishburn (1984))...” (Page 4). Since [1] was officially published at ICML 2024 just four months ago, it is unlikely that the term “Minimax Winner” is as “well-established” as the reviewer stated. We have confirmed that none of the previous work cited by [1] in the quoted sentence explicitly defined or used the term “Minimax Winner.” That said, we are open to considering this term or similar ones. As stated earlier, our primary goal is to use a term or definition that clearly and easily conveys the information.
> >
> > Reference
> >
> > [1] Swamy, Gokul, et al. "A Minimaximalist Approach to Reinforcement Learning from Human Feedback." Forty-first International Conference on Machine Learning.

---

> > ### Author Response · Authors · 2024-11-24
> > **Response to Reviewer tugW’s comment [4/6]**
> >
> > > “2. As the authors stated, the conceptual prox algorithm is proposed for solving variational inequality problems. And this method serves as a general framework for addressing various issues. Introducing this method in the context of alignment does not offer any new insights specific to the alignment problem, nor does it inherently imply the desired property of convergence to the exact Nash equilibrium. Presenting this method only confuses and misleads readers about the actual mechanisms necessary for achieving exact NE convergence—mechanisms that have already been developed and rigorously detailed in [2][3]. Additionally, the proof of COMAL does not introduce any new techniques beyond those already established in [2][3], further emphasizing the lack of novel contributions in this regard.”
> >
> > 1. As emphasized in our previous response, the algorithms proposed in both [2] and [3] closely resemble the conceptual prox algorithms proposed in [1]. Although [1] was not explicitly cited by [2, 3], the algorithms proposed in both [2] and [3] can be viewed as variants of the conceptual prox algorithms proposed in [1] instantiated with different regularizers. We did not claim to have originally proposed the conceptual prox algorithm [1] and have provided proper citations to it in Section 3.1, where our algorithm is introduced, right before we provided references and discussions to [2, 3].
> >
> > 2. We disagree with the assertion that “introducing this method in the context of alignment does not offer any new insights specific to the alignment problem, nor does it inherently imply the desired property of convergence to the exact Nash equilibrium. Presenting this method only confuses and misleads readers …”.
> > First, the conceptual prox method does directly imply last-iterate convergence to an exact Nash equilibirum, as NE are exactly the solutions for the corresponding variaitonal inequality problem. Second, in stark contrast to the comment, we believe introducing the well-established conceptual prox method [1] from the optimization community to alignment is our major contribution and offers new insights into the alignment problem. Despite having been well-established in the optimization community, all the existing work for alignment overlooked the last-iterate convergent conceptual prox method and used the divergent mirror descent method.
> >
> > 3. The major challenge of applying optimization methods to alignment is how to implement them in a practical way for LLM fine-tuning. Only after we unify existing methods through the lens of the prox operator will it become clear that (1) many preference optimization algorithms can be extended to effectively implement the Prox operator; (2) and we can implement several additional optimization methods beyond mirror descent. These methods include the conceptual prox method [1],  mirror-prox method [1],  and the optimistic multiplicative weights update (OMWU) algorithm [4, 5] (we have presented the implementation details for mirror prox and OMWU in Appendix F). We note that all of these algorithms exhibit last-iterate convergence in zero-sum games. We focus on the conceptual prox method for LLM-based experiments and leave the evaluation of the practical performance of mirror-prox and OMWU for future work . We are the first to introduce last-iterate convergent methods to alignment, provide practical implementations, conduct LLM-based experiments, and demonstrate the empirical success of the conceptual prox method in large-scale experiments. Our work demonstrates that viewing alignment from the general optimization perspective is valuable and effective, which is a solid contribution to the intersection of mathematical optimization and LLM alignment.
> >
> > References
> >
> > [1] Nemirovski, Arkadi. "Prox-method with rate of convergence O (1/t) for variational inequalities with Lipschitz continuous monotone operators and smooth convex-concave saddle point problems." SIAM Journal on Optimization 15.1 (2004): 229-251.
> >
> > [2] Perolat, Julien, et al. "From poincaré recurrence to convergence in imperfect information games: Finding equilibrium via regularization." International Conference on Machine Learning. PMLR, 2021.
> >
> > [3] Abe, Kenshi, et al. "Adaptively Perturbed Mirror Descent for Learning in Games." Forty-first International Conference on Machine Learning.
> >
> > [4] Rakhlin, Sasha, and Karthik Sridharan. "Optimization, learning, and games with predictable sequences." Advances in Neural Information Processing Systems 26 (2013).
> >
> > [5] Syrgkanis, V., Agarwal, A., Luo, H., & Schapire, R. E. (2015). Fast convergence of regularized learning in games. Advances in Neural Information Processing Systems, 28.

---

> > ### Author Response · Authors · 2024-11-24
> > **Response to Reviewer tugW’s comment [6/6]**
> >
> > > “6. Adhering to the standard AlpacaEval pipeline [4] is essential. Specifically, evaluations should be conducted against GPT-4-turbo as baseline, rather than relying on pairwise comparisons against the trained models mentioned in the reply. This approach would provide readers with a more comprehensive understanding of COMAL’s performance on the standard benchmark and enable direct comparisons with other opensource strong baselines featured on the open leaderboard [5].”
> >
> > 1. We respectfully disagree with the assertion that “adhering to the standard AlpacaEval pipeline is essential.” (a) As explained in the original submission (Lines 454 - 456), using the same preference oracle used during training (Llama-3-OffsetBias-8B) instead of GPT-4 allows for more controlled experiments, ensuring that the preference oracle used to evaluate the models is also the one the models are trained to fit. (b) In our previous response, we provided evaluation results using GPT-4(-Turbo) as the evaluator to directly compare models trained from different algorithms, which we believe enables more direct comparisons compared to using GPT-4-Turbo as a baseline. (c) Moreover, as emphasized in our previous response, our focus is on a policy’s win rate against _any_ policies, therefore, we do not consider the win rate against a specific model (GPT-4-Turbo) essential.
> >
> > 2. That said, we do appreciate the reviewer’s suggestion and recognize the value of the suggested evaluation in enabling a more direct comparison with previous work. Accordingly, we have included experiments using GPT-4-Turbo (gpt-4-1106-preview) as both the baseline system and the evaluator on AlpacaEval – the default setting of AlpacaEval2. The results demonstrate that the Qwen-2-1.5B checkpoint, trained with COMAL, not only outperformed other iterative preference optimization algorithms but also some 7B LLMs. These include results from the official AlpacaEval 2 leaderboard: llama2-chat-7B and Alpaca 7B. This further addresses the reviewer’s previous comment that “the model used, Qwen2-1.5B, is too small to be meaningful – at least a 7B parameter model should have been used to demonstrate effectiveness.” We will include this result in the revised version.
> >
> > | Model/Method | AlpacaEval2 Length-Controlled Score |
> > |--------------|-------------------------------------|
> > | Llama-2-7B-Chat      | 5.4                           |
> > | Alpaca 7B      | 5.9                             |
> > | SFT      | 4.8                             |
> > | Iter-IPO      | 4.8                           |
> > | INPO (Round 3)     | 5.6                             |
> > | COMAL     | 6.2                             |
> >
> > References
> >
> > [4] Dubois, Yann, et al. "Length-controlled alpacaeval: A simple way to debias automatic evaluators." arXiv preprint arXiv:2404.04475 (2024).
> >
> > [5] https://tatsu-lab.github.io/alpaca_eval/

---

> > > ### Comment · Reviewer_tugW · 2024-12-02
> > >
> > > Thanks for the detailed response! It's nice to see the authors revise their manuscript to improve the paper. However, after careful consideration based on previous discussion, I have decided to maintain my original score. Here are the reasons for this decision:
> > > 1. While the authors claim that introducing the term "robust alignment" is not a major contribution of this work, the framing and experimental design of the paper are heavily centered around this concept. In particular, the main empirical results aim to demonstrate that the trained policy achieves "robust alignment," making it a central theme of the work. However, this concept is problematic both in theory and practice. As noted in previous discussions, a policy achieving robust alignment is essentially equivalent to a Nash equilibrium policy. Therefore, introducing a new term for this concept seems unnecessary and adds complexity to the paper, making it harder for readers to understand, especially those who are unfamiliar with game theory. Although the authors suggest that the term simplifies discussions, it instead increases the cognitive burden on readers. Moreover, I do not see how this concept distinguishes the Nash equilibrium of the modified game from that of the original game. Furthermore, the definition of "robust alignment"—ensuring a minimum 50% win rate against any competing strategy—feels arbitrary and lacks substantive meaning. For example, in the supplementary experiments, COMAL achieves only a 6.2% win rate against GPT-4 Turbo, which significantly undermines the practical value of this concept. I have never encountered any paper that defines Nash equilibrium in this manner or uses such a definition as a basis for main results and validation. Theoretically, in game theory, exploitability is the standard metric for evaluating the gap between a policy and a Nash equilibrium. While I understand that computing exploitability might be challenging in alignment tasks, it is crucial for COMAL to demonstrate strong empirical performance on standard benchmark.
> > > 2. The Minimax Winner (or Von Neumann Winner) [1][2] is a well-defined and widely recognized solution concept. It is deeply rooted in social choice theory [3][4] and has been extensively discussed in key literature on game-theoretical approaches to alignment [2][5][6][7]. This concept not only has solid theoretical foundations but also offers a clear practical interpretation—never pick a solution that makes a significant portion of the population consistently unhappy. It's surprising that the authors are unfamiliar with this fundamental solution concept despite following its formulation. I would like to kindly remind the authors that it is their responsibility to thoroughly review the foundational literature in the field. A strong understanding of why game-theoretical approaches are introduced, what the solution concepts represent, and how they apply to alignment is essential. Verifying the applicability and significance of the Minimax Winner would provide much more meaningful contributions than proposing a new concept like "robust alignment."
> > > 3. I remain unconvinced that incorporating conceptual proximal methods constitutes a novel contribution. As far as I can tell, the proofs of main theorems of COMAL heavily rely on well-established methods from prior works [8][9], without introducing new techniques. As mentioned earlier, conceptual proximal methods are general techniques applicable across various problems. To claim this as a major contribution is akin to stating that English literature, French literature, and German literature all fall under the category of literature. Additionally, OMWU does not directly apply to COMAL's framework, as OMWU itself achieves last-iterate convergence to the exact Nash equilibrium. COMAL is also not the first work to introduce last-iterate convergence concept to the alignment field (see "Convergence of the last iterate" part of Section 5 in [10], footnote of page 6 and page 22 in [2]).
> > > 4. I did not deny that INPO can achieve last-iterate convergence to the NE of the modified game. What I mean is that COMAL only runs a single iteration of INPO (i.e., $\pi_t$  to $\pi_{t+1}$), which corresponds to a best response. However, INPO requires $T$ iterations (where $t=0,1,2,...,T$) to converge to the NE of the modified game. To better illustrate this point, consider the supplementary results on AlpacaEval: how many mirror descent iterations of INPO did the authors run? At what point did the authors replace the reference policy?

---

> > > > ### Comment · Reviewer_tugW · 2024-12-02
> > > >
> > > > 5. The authors need to clearly state in the main text that the original forms of the SPPO loss, the DRO loss, and the REBEL loss do not directly apply to COMAL, to avoid overstating its generalization capability. For the adapted versions, if the authors claim that these versions can work within the COMAL framework, they must provide comprehensive experimental comparisons to substantiate these claims. Simply presenting results from a modified version of INPO is insufficient to validate the claimed generalization. Additionally, Section 4 is not suitable for main results. It would be better to use it as a motivational example. Section 5 should present the standard benchmark results and baseline comparisons for all of the proposed adaptations, including the adapted versions of SPPO, DRO, REBEL, INPO, and OMWU, since COMAL claims that these adaptations are all suitable for integration into the framework.
> > > >
> > > > References
> > > >
> > > > [1] Dudík, M., Hofmann, K., Schapire, R. E., Slivkins, A., & Zoghi, M. (2015, June). Contextual dueling bandits. In Conference on Learning Theory (pp. 563-587). PMLR.
> > > >
> > > > [2] Swamy, Gokul, et al. "A minimaximalist approach to reinforcement learning from human feedback." arXiv preprint arXiv:2401.04056 (2024).
> > > >
> > > > [3] Kreweras, G. Aggregation of preference orderings. In Mathematics and Social Sciences I: Proceedings of the seminars of Menthon-Saint-Bernard, France (1–27 July 1960) and of Gösing, Austria (3–27 July 1962), pp. 73–79, 1965.
> > > >
> > > > [4] Lanctot, M., Larson, K., Bachrach, Y., Marris, L., Li, Z., Bhoopchand, A., ... & Koop, A. (2023). Evaluating Agents using Social Choice Theory. arXiv preprint arXiv:2312.03121.
> > > >
> > > > [5] Ye, Chenlu, et al. "Online iterative reinforcement learning from human feedback with general preference model." The Thirty-eighth Annual Conference on Neural Information Processing Systems. 2024.
> > > >
> > > > [6] Rosset, Corby, et al. "Direct nash optimization: Teaching language models to self-improve with general preferences." arXiv preprint arXiv:2404.03715 (2024).
> > > >
> > > > [7] Wu, Yue, et al. "Self-play preference optimization for language model alignment." arXiv preprint arXiv:2405.00675 (2024).
> > > >
> > > > [8] Perolat, Julien, et al. "From poincaré recurrence to convergence in imperfect information games: Finding equilibrium via regularization." International Conference on Machine Learning. PMLR, 2021.
> > > >
> > > > [9] Abe, Kenshi, et al. "Adaptively Perturbed Mirror Descent for Learning in Games." Forty-first International Conference on Machine Learning.
> > > >
> > > > [10] Munos, R., Valko, M., Calandriello, D., Azar, M. G., Rowland, M., Guo, Z. D., ... & Piot, B. (2023). Nash learning from human feedback. arXiv preprint arXiv:2312.00886.

---

> > > > > ### Author Response · Authors · 2024-12-03
> > > > > **Response to Reviewer tugW’s additional comments [1/5]**
> > > > >
> > > > > We thank the reviewer for taking the time to provide additional feedback. However, it appears that the reviewer did not fully consider our previous responses, as there are still factual misunderstandings of our work in the new comments. There are also questions with clear answers in our original manuscript. Additionally, there are logical inconsistencies in the response, which we will further elaborate on. As we find ourselves reiterating our previous points and clarifying misunderstandings, we once again kindly request the reviewer to carefully and objectively assess our submission and our responses.
> > > > >
> > > > > We would like to begin with a general response to the first and the second reasons to reject provided in the reviewer’s new comment.
> > > > >
> > > > > 1. **Logical inconsistencies in the first two reasons provided by the reviewer.**
> > > > >
> > > > > To demonstrate these inconsistencies, we will first provide a discussion of the von Neumann winner [1]. We note that the notion of von Neumann winner [1] is defined as the policy with the robust alignment property that achieves at least a 50\% win rate against any other policies. We quote section 2 from [1] here:
> > > > > >“In words, for every action $b$, if $a$ is selected randomly according to distribution $w$, then the chance of beating $b$ in a duel is at least 1/2. (Note that this property implies that the same will be true if $b$ is itself selected in a randomized way.) A distribution $w$ with this property is said to be a von Neumann winner for the preference matrix $P$.”
> > > > >
> > > > > [1] does not give this property (the chance of beating any randomized strategy in a duel is at least 1/2) a name, but this property is precisely what we refer to in the paper as “robust alignment”: at least a 50% win rate against any other policy (Lines 51-52). We thank the reviewer for providing the reference [1] and will add the reference and comment when we introduce “robust alignment.” We remark again (as in previous responses) that “robust alignment” is not a contribution but just a notation (tailored for readers from the LLM alignment community who may not be familiar with game theory). That said, we are willing to replace all “robust alignment” with its definition: achieving at least 50% win rate against any other policy. However, as discussed in the previous response, keeping “robust alignment” simplifies the notation.
> > > > >
> > > > > Given that the von Neumann Winner [1] is defined as the policy with robust alignment, aka a 50% win rate against any other policy, we found the reviewer’s comments inconsistent, resulting from potential misunderstandings of the definitions of “robust alignment” and “von Neumann winner.” The inconsistent claims are as follows.
> > > > >
> > > > > P1: “However, this concept (robust alignment) is problematic in theory and practice”
> > > > >
> > > > > P2: “Furthermore, the definition of "robust alignment"—ensuring a minimum 50% win rate against any competing strategy—feels arbitrary and lacks substantive meaning. ”
> > > > >
> > > > > P3:  “I have never encountered any paper that defines Nash equilibrium in this manner or uses such a definition (robust alignment) as a basis for main results and validation.”
> > > > >
> > > > > P4: “The Minimax Winner (or Von Neumann Winner) [1][2] is a well-defined and widely recognized solution concept. It is deeply rooted in social choice theory [3][4] and has been extensively discussed in key literature on game-theoretical approaches to alignment [2][5][6][7]. This concept not only has solid theoretical foundations but also offers a clear practical interpretation”
> > > > >
> > > > > As clarified at the beginning, the Von Neumann Winner [1] is defined precisely as the policy with the robust alignment property, aka a 50% win rate against any other policy. Thus, **claims P1, P2, and P3 about our paper are incorrect and inconsistent with P4.**
> > > > >
> > > > > References
> > > > >
> > > > > [1] Dudík, M., Hofmann, K., Schapire, R. E., Slivkins, A., & Zoghi, M. (2015, June). Contextual dueling bandits. In Conference on Learning Theory (pp. 563-587). PMLR.
> > > > >
> > > > >
> > > > > [2] Swamy, Gokul, et al. "A minimaximalist approach to reinforcement learning from human feedback." arXiv preprint arXiv:2401.04056 (2024).
> > > > >
> > > > >
> > > > > [3] Kreweras, G. Aggregation of preference orderings. In Mathematics and Social Sciences I: Proceedings of the seminars of Menthon-Saint-Bernard, France (1–27 July 1960) and of Gösing, Austria (3–27 July 1962), pp. 73–79, 1965.
> > > > >
> > > > >
> > > > > [4] Lanctot, M., Larson, K., Bachrach, Y., Marris, L., Li, Z., Bhoopchand, A., ... & Koop, A. (2023). Evaluating Agents using Social Choice Theory. arXiv preprint arXiv:2312.03121.
> > > > >
> > > > >
> > > > > [5] Ye, Chenlu, et al. "Online iterative reinforcement learning from human feedback with general preference model." The Thirty-eighth Annual Conference on Neural Information Processing Systems. 2024.
> > > > >
> > > > >
> > > > > [6] Rosset, Corby, et al. "Direct nash optimization: Teaching language models to self-improve with general preferences." arXiv preprint arXiv:2404.03715 (2024).
> > > > >
> > > > >
> > > > > [7] Wu, Yue, et al. "Self-play preference optimization for language model alignment." arXiv preprint arXiv:2405.00675 (2024).

---

> > > > > ### Author Response · Authors · 2024-12-03
> > > > > **Response to Reviewer tugW’s additional comments [2/5]**
> > > > >
> > > > > 2. **Additional responses regarding the reviewer’s first two reasons to reject.**
> > > > >
> > > > > > P5: “Moreover, I do not see how this concept distinguishes the Nash equilibrium of the modified game from that of the original game.”
> > > > >
> > > > > A: We have clarified in the previous response: (1) the Nash equilibrium of the modified game does not achieve robust alignment (50% win rate against any other policy); (2) the Nash equilibrium of the original game achieves robust alignment (50% win rate against any other policy). Thus, robust alignment distinguishes the Nash equilibrium of the modified game from that of the original game. Please let us know if the above clarification is clear.
> > > > >
> > > > > ----
> > > > >
> > > > > > P6: “I have never encountered any paper that defines Nash equilibrium in this manner or uses such a definition as a basis for main results and validation ... in the supplementary experiments, COMAL achieves only a 6.2% win rate against GPT-4 Turbo, which significantly undermines the practical value of this concept.”
> > > > >
> > > > > A: We disagree with this point, which contains misunderstandings of both our paper and previous works. We present the reasons below.
> > > > >
> > > > > 1. We define Nash equilibrium as “$ \arg\max_{\pi_1 \in \Pi} \arg\min_{\pi_2 \in \Pi} J(\pi_1,\pi_2)$ ” in Lines 139-140, which is consistent with all the previous works. We note that robust alignment is only a property (not the definition) of the Nash equilibrium, as stated in Line 52 and Lines 141-143.
> > > > > 2. Finding the Nash equilibrium policy of the unregularized game is the basis of our main results, which is clearly stated in Theorem 1 and throughout the paper. Achieving robust alignment is a byproduct of the Nash equilibrium policy.
> > > > > 3. Using a pairwise win rate for experimental illustration is standard in previous works. The work on Nash learning in LLM alignment [10] conducts experiments showing that the proposed MD1 algorithm has more than a 50% win rate against any other baseline methods. This is because achieving a 50% win rate against any other policy is a property of the Nash equilibrium policy and is relatively easy to illustrate among models with similar sizes. We remark that not achieving 50% against GPT-4 is expected, given the significant difference in the model size. This does not undermine the value of this concept since the von Neumann winner is by definition a policy with the robust alignment property. Furthermore, the provided additional evaluation results show that COMAL achieves a strictly greater than 50% win rate against all baseline algorithms when training from the same base SFT policy, showing its superiority over other iterative algorithms in approximating the Nash equilibrium.
> > > > >
> > > > > Reference
> > > > >
> > > > > [10] Munos, R., Valko, M., Calandriello, D., Azar, M. G., Rowland, M., Guo, Z. D., ... & Piot, B. (2023). Nash learning from human feedback. arXiv preprint arXiv:2312.00886.

---

> > > > > ### Author Response · Authors · 2024-12-03
> > > > > **Response to Reviewer tugW’s additional comments [3/5]**
> > > > >
> > > > > 2. **Additional responses regarding the reviewer’s first two reasons to reject (continue).**
> > > > >
> > > > > > P7: “The Minimax Winner (or Von Neumann Winner) [1][2] is a well-defined and widely recognized solution concept. It is deeply rooted in social choice theory [3][4] and has been extensively discussed in key literature on game-theoretical approaches to alignment [2][5][6][7]. This concept not only has solid theoretical foundations but also offers a clear practical interpretation—never pick a solution that makes a significant portion of the population consistently unhappy. It's surprising that the authors are unfamiliar with this fundamental solution concept despite following its formulation. I would like to kindly remind the authors that it is their responsibility to thoroughly review the foundational literature in the field. A strong understanding of why game-theoretical approaches are introduced, what the solution concepts represent, and how they apply to alignment is essential. Verifying the applicability and significance of the Minimax Winner would provide much more meaningful contributions than proposing a new concept like "robust alignment."”
> > > > >
> > > > > A:
> > > > > 1. Please see our previous response for a general discussion regarding the connection between the Von Neumann Winner and “robust alignment”.
> > > > >
> > > > > 2. We are aware of the concept of “Von Neumann Winner” and provided related discussions in our manuscript in Lines 929 - 931: “Munos et al. (2024)[10] also consider general preferences and aim to find the von Neumann winner, which corresponds to the Nash equilibrium of a game played between the two LLMs over the win rate.”
> > > > >
> > > > > 3. As clearly stated in our previous response, the alternative term “Minimax Winner” strongly suggested by the reviewer seems to be defined in the reference they provided [2]: “we define the Minimax Winner (MW, Kreweras (1965) [3]; Simpson (1969); Kramer (1973); Fishburn (1984))...” (Page 4). Since [2] was officially published at ICML 2024 just four months ago, it is unlikely that the term “Minimax Winner” is as “well-established” as the reviewer stated. We have confirmed that none of the previous work cited by [2] in the quoted sentence, including [3], explicitly defined or used the term “Minimax Winner.” This term is also not explicitly defined or used in [1][4][7]. [5][6] did mention “Minimax Winner” when discussing the background of their methods. However, both of them cited [2] and provided a very similar discussion and the same references for “Minimax Winner”. As stated before, we have verified that none of the previous references cited by [2] (and by [5][6]), explicitly defined or used the term “Minimax Winner”. Therefore, we found the strong claim made by the reviewer regarding our lack of understanding of “the foundational literature in the field” ungrounded.
> > > > >
> > > > >
> > > > > References
> > > > >
> > > > >
> > > > > [1] Dudík, M., Hofmann, K., Schapire, R. E., Slivkins, A., & Zoghi, M. (2015, June). Contextual dueling bandits. In Conference on Learning Theory (pp. 563-587). PMLR.
> > > > >
> > > > > [2] Swamy, Gokul, et al. "A minimaximalist approach to reinforcement learning from human feedback." arXiv preprint arXiv:2401.04056 (2024).
> > > > >
> > > > > [3] Kreweras, G. Aggregation of preference orderings. In Mathematics and Social Sciences I: Proceedings of the seminars of Menthon-Saint-Bernard, France (1–27 July 1960) and of Gösing, Austria (3–27 July 1962), pp. 73–79, 1965.
> > > > >
> > > > > [4] Lanctot, M., Larson, K., Bachrach, Y., Marris, L., Li, Z., Bhoopchand, A., ... & Koop, A. (2023). Evaluating Agents using Social Choice Theory. arXiv preprint arXiv:2312.03121.
> > > > >
> > > > > [5] Ye, Chenlu, et al. "Online iterative reinforcement learning from human feedback with general preference model." The Thirty-eighth Annual Conference on Neural Information Processing Systems. 2024.
> > > > >
> > > > > [6] Rosset, Corby, et al. "Direct nash optimization: Teaching language models to self-improve with general preferences." arXiv preprint arXiv:2404.03715 (2024).
> > > > >
> > > > > [7] Wu, Yue, et al. "Self-play preference optimization for language model alignment." arXiv preprint arXiv:2405.00675 (2024).
> > > > >
> > > > > [10] Munos, R., Valko, M., Calandriello, D., Azar, M. G., Rowland, M., Guo, Z. D., ... & Piot, B. (2023). Nash learning from human feedback. arXiv preprint arXiv:2312.00886.

---

> > > > > ### Author Response · Authors · 2024-12-03
> > > > > **Response to Reviewer tugW’s additional comments [4/5]**
> > > > >
> > > > > **Responses to Reviewer’s other reasons to reject.**
> > > > >
> > > > > > “I remain unconvinced that incorporating conceptual proximal methods constitutes a novel contribution. As far as I can tell, the proofs of main theorems of COMAL heavily rely on well-established methods from prior works [8][9], without introducing new techniques. As mentioned earlier, conceptual proximal methods are general techniques applicable across various problems. To claim this as a major contribution is akin to stating that English literature, French literature, and German literature all fall under the category of literature.”
> > > > >
> > > > > A: We note that while our proof is inspired by [11] and the subsequent works [8][9], introducing the conceptual prox method to LLM alignment and conducting LLM-based experiments require more insights than the works [8][9], which only concerns the (small-scale) tabular setting and is not practical for the LLM setting. As we have clarified in the previous response, the major challenge is how to practically implement the prox operator in large-scale settings. We make the novel observation that many existing alignment methods can be extended to compute the prox operator and are the first to (1) give practical implementation of last-iterate convergent methods (for unregularized game) including the conceptual prox, OMWU, and mirror-prox; (2) conduct LLM-based experiments for the conceptual prox (COMAL) and show its effectiveness over non-convergent baselines. This is a solid contribution bridging optimization, game theory, and LLM alignment.
> > > > >
> > > > > ----
> > > > >
> > > > > >“Additionally, OMWU does not directly apply to COMAL's framework, as OMWU itself achieves last-iterate convergence to the exact Nash equilibrium.”
> > > > >
> > > > > A: We do not see why this is a reason for rejection. We never claimed that OMWU could / or even should be applied to COMAL. As stated in the previous response and Appendix F, by unifying existing alignment losses through the lens of the prox operator, we give practical implementations of several other optimization methods including mirror-prox and OMWU that also has last-iterate convergence. Our contribution here is providing the first practical implementation of OMWU and mirror-prox in the context of LLM alignment.
> > > > >
> > > > >
> > > > >
> > > > > ----
> > > > >
> > > > > > “COMAL is also not the first work to introduce last-iterate convergence concept to the alignment field (see "Convergence of the last iterate" part of Section 5 in [10], footnote of page 6 and page 22 in [2]).”
> > > > >
> > > > > 1. We note that the statement in our (revised) manuscript and our previous response is “our work is the first to introduce last-iterate convergent _methods_ to LLM alignment” (Lines 972 - 973). We acknowledge that there is an underspecification in this statement as we were mainly discussing the unregularized (unmodified) games. A more accurate statement would be “our work is the first to introduce last-iterate convergent methods **of the unregularized games** to LLM alignment”. We will make this change in the revised version.
> > > > >
> > > > > 2. We’d like to note that we have already provided discussions of [10][2] in our original manuscript. Specifically, in Table 1 and Line 148, we clearly stated that the algorithm proposed in [10] converges to the **modified KL-regularized game**. Moreover, in Lines 930 - 932, we stated “they [10] propose a variant of the Mirror Descent (MD) algorithm called Nash-MD and show last-iterate convergence in the KL-regularized game“. Regarding [2], we provide discussions in Lines 933 - 934: “concurrently, [2] study the same solution concept focusing more on sequential games.” We note that the authors of [2] only briefly stated the possibility of extending their algorithm to achieve last-iterate convergence without providing a thorough discussion or empirical experiments.
> > > > >
> > > > > References
> > > > >
> > > > > [2] Swamy, Gokul, et al. "A minimaximalist approach to reinforcement learning from human feedback." arXiv preprint arXiv:2401.04056 (2024).
> > > > >
> > > > > [8] Perolat, Julien, et al. "From poincaré recurrence to convergence in imperfect information games: Finding equilibrium via regularization." International Conference on Machine Learning. PMLR, 2021.
> > > > >
> > > > > [9] Abe, Kenshi, et al. "Adaptively Perturbed Mirror Descent for Learning in Games." Forty-first International Conference on Machine Learning.
> > > > >
> > > > > [10] Munos, R., Valko, M., Calandriello, D., Azar, M. G., Rowland, M., Guo, Z. D., ... & Piot, B. (2023). Nash learning from human feedback. arXiv preprint arXiv:2312.00886.
> > > > >
> > > > > [11] Nemirovski, Arkadi. "Prox-method with rate of convergence O (1/t) for variational inequalities with Lipschitz continuous monotone operators and smooth convex-concave saddle point problems." SIAM Journal on Optimization 15.1 (2004): 229-251.

---

> > > > > ### Author Response · Authors · 2024-12-03
> > > > > **Response to Reviewer tugW’s additional comments [5/5]**
> > > > >
> > > > > **Responses to Reviewer’s other reasons to reject (continue).**
> > > > >
> > > > > > “I did not deny that INPO can achieve last-iterate convergence to the NE of the modified game. What I mean is that COMAL only runs a single iteration of INPO (i.e., $\pi_t$ to $\pi_{t+1}$), which corresponds to a best response. However, INPO requires $T$ iterations (where $t=0,1,2,...,T$) to converge to the NE of the modified game. To better illustrate this point, consider the supplementary results on AlpacaEval: how many mirror descent iterations of INPO did the authors run? At what point did the authors replace the reference policy?”
> > > > >
> > > > > 1. *“What I mean is that COMAL only runs a single iteration of INPO.”* A: We believe our previous response has clearly addressed this misunderstanding. As stated clearly in Algorithm 3, INPO is an iterative solver for the regularized Nash equilibrium of the regularized game $J_{\tau}(\pi_1, \pi_2, \pi_{ref})$: it performs iterations of mirror descent over the regularized game $J_{\tau}(\pi_1, \pi_2, \pi_{ref})$ and returns an approximate regularized Nash equilibrium. Line 345 in Algorithm 4 sets $\pi^{t+1}$ to be the output of $INPO(\pi_{ref}, \tau_t, \eta_t, K_t, \mathbb{P})$ which is the regularized Nash equilibrium returned by running $K_t$ iterations of INPO. Thus, Line 345 in **Algorithm 4 corresponds to $K_t > 1$ iterations (instead of a single iteration) of mirror descent** over the regularized game.
> > > > >
> > > > > 2. To provide further clarification in case there is still any misunderstanding, we note that COMAL is an iterative algorithm that applies the INPO *algorithm* as its subroutine. INPO *algorithm* itself is also an iterative algorithm that iteratively uses the INPO *loss* to update the policy.
> > > > >
> > > > > 3. *“how many mirror descent iterations of INPO did the authors run? At what point did the authors replace the reference policy?”* A: There are clear answers to these questions in the main text of our original and updated manuscripts. Specifically, it is described in Lines 516 - 518: *“for COMAL, which shares the same training trajectory as INPO for the first two training rounds, we update the reference policy at the beginning of the third training round, following the optimization process described in Algorithm 4”*. Lines 482 - 484 state *“for INPO and COMAL, the model is trained for up to 18 iterations, which are equivalent to 3 training rounds over the entire instruction set since it has been split into 6 subsets.”* Therefore, two training rounds are equivalent to 12 iterations in INPO. We note that under this controlled experiment where the cost of computing is the same, COMAL outperforms INPO after both 2 and 3 rounds of training (Table 2).
> > > > >
> > > > > - - - -
> > > > >
> > > > > >“The authors need to clearly state in the main text that the original forms of the SPPO loss, the DRO loss, and the REBEL loss do not directly apply to COMAL, to avoid overstating its generalization capability. For the adapted versions, if the authors claim that these versions can work within the COMAL framework, they must provide comprehensive experimental comparisons to substantiate these claims. Simply presenting results from a modified version of INPO is insufficient to validate the claimed generalization.”
> > > > >
> > > > > A: We thank the reviewer for acknowledging that the SPPO loss, the DRO loss, and the REBEL can be extended and then applied to COMAL. We will make it clear that modification is necessary to extend these losses from unregularized preference to regularized preferences, and our extensions might be of independent interest for future works. We have provided LLM-based experiments with COMAL with the INPO loss and shown its strong performance than other baseline methods; but we agree that it is interesting to test the performances of COMAL integrated with other losses and leave it as future works.

---

> ### Author Response · Authors · 2024-11-24
> **Response to Reviewer tugW’s comment [1/6]**
>
> We thank the reviewer for their additional feedback. We believe that most of the points in the reviewer’s new response, which largely reiterate the original review, have already been addressed in our previous response. We would appreciate it if the reviewer could give our original response more careful consideration. Below, we provide further clarification.
>
> First, we would like to address two critical concerns raised by the reviewer's recent comments. We will restate the two propositions that the reviewer has concerns about, and provide explanations as we already did in our submission and the original response. We would appreciate it if the reviewer could re-evaluate these propositions, taking our new response into consideration, and provide a clear response: whether the reviewer agrees that these propositions are valid or not. If the reviewer insists the propositions are not valid, we would appreciate further explanation or evidence to support the reviewer’s opinion.
>
> **Proposition 1: The practical implementation of Algorithm 4 is consistent with Algorithm 1.**
>
> The reviewer’s comment regarding Proposition 1:
> > “4. While Algorithm 1 (Line 196) is conceptually correct, the iterative process described in the practical implementation of Algorithm 4 is inconsistent with Algorithm 1, and from my perspective, the differences are significant. Algorithm 1 suggests that to converge to the exact Nash equilibrium (NE), it is necessary first to converge to the NE of the KL-regularized game and then update the reference policy to this NE. However, in line 345 of Algorithm 4, the opponent is fixed, and INPO only learns a single best response to the opponent. This learned policy represents a best response rather than the NE of the KL-regularized game. Moreover, line 345 in Algorithm 4 corresponds to a single iteration of mirror descent (i.e.,
> , not the gradient update step in the authors’ reply). Whether this single iteration can produce a meaningful response is highly questionable, let alone achieve convergence to the NE of the KL-regularized game. And the approximation error is not studied. This significant inconsistency between Algorithm 1 and Algorithm 4 is misleading, making the implementation in Algorithm 4 lack solid theoretical foundations.”
>
> A: We disagree with the assertions made in the comment, and provide further clarifications below:
>
> 1. The claim that “INPO only learns a single best response to the opponent. This learned policy represents a best response rather than the NE of the KL-regularized game” is incorrect and a misunderstanding of INPO (Algorithm 3) proposed in [1]. It seems the reviewer thinks we should update the policies of both players in the game. This is correct if the game is asymmetric. However, **the alignment game is symmetric, and thus, the updates of both players are the same** (which has been observed and used in all recent works on the topic (e.g., [1,2,3,4])]). As a result, we only need to update the policy $\mu^k$ for one player. After sufficient iterations, $(\mu^K, \mu^K)$ converges to the NE of the regularized game, for which we show exponentially fast convergence in Theorem 2. Thus, **INPO (Algorithm 3), the subroutine of Algorithm 4, converges to a regularized Nash equilibrium.**
>
> 2. The claim “Moreover, line 345 in Algorithm 4 corresponds to a single iteration of mirror descent” is incorrect and a misunderstanding. As stated clearly in Algorithm 3, INPO is an iterative solver for the regularized Nash equilibrium of the regularized game $J_{\tau}(\pi_1, \pi_2, \pi_{ref})$: it performs iterations of mirror descent over the regularized game $J_{\tau}(\pi_1, \pi_2, \pi_{ref})$ and returns an approximate regularized Nash equilibrium. Line 345 in Algorithm 4 sets $\pi^{t+1}$ to be the output of $INPO(\pi_{ref}, \tau_t, \eta_t, K_t, \mathbb{P})$ which is the regularized Nash equilibrium returned by running $K_t$ iterations of INPO. Thus, Line 345 in **Algorithm 4 corresponds to $K_t > 1$ iterations (instead of a single iteration) of mirror descent** over the regularized game. Since Line 345 in Algorithm 4 is INPO, the regularized game solver for $J_\tau(\pi_1,\pi_2, \pi_{ref})$, **there is no inconsistency between Algorithm 4 and Algorithm 1, and Algorithm 4 is a practical implementation of Algorithm 1.**
>
> References
>
> [1] Zhang, Yuheng, et al. "Iterative Nash Policy Optimization: Aligning LLMs with General Preferences via No-Regret Learning." arXiv preprint arXiv:2407.00617 (2024).
>
> [2] Swamy, Gokul, et al. "A minimaximalist approach to reinforcement learning from human feedback." arXiv preprint arXiv:2401.04056 (2024).
>
> [3] Munos, Remi, et al. "Nash Learning from Human Feedback." Forty-first International Conference on Machine Learning. 2024
>
> [4] Wu, Yue, et al. "Self-play preference optimization for language model alignment." arXiv preprint arXiv:2405.00675 (2024).

---

> ### Author Response · Authors · 2024-11-24
> **Response to Reviewer tugW’s comment [5/6]**
>
> > “3. As noted in the second point, the core idea of this paper is largely derived from recent results in [2][3], particularly the results in [3]. However, in the ‘Last-Iterate Convergence on Games’ section of the Related Work, the authors fail to mention or compare their work with these two most closely related studies. This omission is particularly surprising given that, as the authors highlighted in the reply, in lines 180–184, the authors acknowledge the importance of these works. And Lemma 4 in the appendix is directly adapted from Lemma F.1 in [3]. Such a lack of engagement with directly relevant prior research undermines the contextual rigor and completeness of the paper.”
>
> 1. “As noted in the second point, the core idea of this paper is largely derived from recent results in [2][3], particularly the results in [3].” A: As highlighted in our original submission, our algorithm is inspired by the conceptual prox-method [1] (Lines 186 - 187). Please see our previous response regarding the connection between [1] and [2,3].
>
> 2. “And Lemma 4 in the appendix is directly adapted from Lemma F.1 in [3]. Such a lack of engagement with directly relevant prior research undermines the contextual rigor and completeness of the paper.” A: The first quoted sentence exactly reflects the information conveyed in Line 1026 of our submission: "Lemma 4 (Adapted from Lemma F.1 in Abe et al. (2024))." It is difficult to see how this constitutes "a lack of engagement with" relevant prior work.
>
> 3. As noted in our previous response and acknowledged by the reviewer, we properly cited references [2, 3] in the main text of our original submission (Lines 180 - 184) in Section 3.1, which is an important section of our submission, right before introducing our algorithm. Therefore, we disagree with the claim that there is “a lack of engagement with directly relevant prior research.” We recognize that we did not discuss these works further in the related work section in the appendix, given their discussion in a salient place in the main text. In our previous response, we also acknowledged the reviewer's comment and confirmed that we will include further discussions of these two works in the related work section.
>
>
> References
>
> [1] Nemirovski, Arkadi. "Prox-method with rate of convergence O (1/t) for variational inequalities with Lipschitz continuous monotone operators and smooth convex-concave saddle point problems." SIAM Journal on Optimization 15.1 (2004): 229-251.
>
> [2] Perolat, Julien, et al. "From poincaré recurrence to convergence in imperfect information games: Finding equilibrium via regularization." International Conference on Machine Learning. PMLR, 2021.
>
> [3] Abe, Kenshi, et al. "Adaptively Perturbed Mirror Descent for Learning in Games." Forty-first International Conference on Machine Learning.

---

### Official Review · Reviewer_SZrT · 2024-11-04

**Soundness:** 2
**Presentation:** 2
**Contribution:** 2
**Rating:** 3
**Confidence:** 4

**Summary:**

This work proposes COMAL, which is designed to address the limitations of existing alignment methods for Large Language Models (LLMs), such as Reinforcement Learning from Human Feedback (RLHF). While methods like RLHF assume human preferences can be modeled with the Bradley-Terry (BT) approach, the authors argue that this simplification does not capture the full complexity of general human preferences. To overcome this, the authors frame the alignment as a two-player zero-sum game, with the goal of finding a Nash equilibrium policy that achieves robust alignment.

**Strengths:**

- The paper is written very well and easy to follow.

**Weaknesses:**

- For motivation, the authors have argued that transitivity may not hold in practice, while in theory, it makes sense. However, it would be nice to provide some evidence of such behavior in the real-world preference datasets available in practice.

- Further, to deal with this issue, a general preference optimization framework is utilized, as proposed in existing literature, but this has some issues. While collecting the preferences offline from humans makes sense, it is extremely hard to realize the assumption of having access to preference oracle. Are there any real-world examples of such oracles? For the experiments in this work, an LLM is used for preferences, which are AI-generated only then. Does it mean the framework of general preferences is only realistic with AI feedback not human?

- The contributions are incremental because the authors have just utilized the prox operator, which is well-defined and developed in the optimization theory. Are there any additional challenges in applying that to the alignment context?

- Why iterative averaging is cumbersome: we can easily maintain a running average of the model; this way, we don't need to store all the LLMs generated during the training.

- Why does the alignment objective in (3) make sense because there is no KL regularized term in the objective? The authors have mentioned that the objective without KL can no longer achieve robust alignment. We note that straying closer to the based model via KL is one of the most important properties of any alignment method. It does not makes sense to solve a problem without it.

- The proposed algorithm again solves a regularized problem only in the algorithm; it is confusing and hard to follow what is the exact approach. I see the authors have proposed to change the reference model to new pi^{t+1}, but this is weird because the final optimal model can be too far from the based model SFT, which is not required at all. Note that the alignment problem is just not only to maximize the preference provided by the preference oracle, but it is also important to stay closer to the base SFT model.

- How win rate is calculated? Is it via GPT-4 ?

- Also, the win rate plot without showing KL divergence to the SFT model is an incomplete description of the alignment performance of the approach. See the following alignment papers for that:

 Mudgal S, Lee J, Ganapathy H, Li Y, Wang T, Huang Y, Chen Z, Cheng HT, Collins M, Strohman T, Chen J. Controlled decoding from language models. arXiv preprint arXiv:2310.17022. 2023 Oct 25.

Chakraborty S, Ghosal SS, Yin M, Manocha D, Wang M, Bedi AS, Huang F. Transfer Q Star: Principled Decoding for LLM Alignment. arXiv preprint arXiv:2405.20495. 2024 May 30.

**Questions:**

Please see the weakness.

---

> ### Author Response · Authors · 2024-11-21
> **Response to Reviewer SZrT [1/4]**
>
> **General response**
>
> We appreciate the review. We would like to begin by addressing a few points that may have been misunderstood.
>
> > ”Why iterative averaging is cumbersome: we can easily maintain a running average of the model; this way, we don't need to store all the LLMs generated during the training.”
>
> Maintaining a running average of model parameters fails to achieve the required iterative averaging, which should instead be applied to the models’ outputs, not their parameters (Lines 155-156). Specifically, let $\theta^t$ denote the parameter of the $t$-th model. We note that output from $\frac{1}{T} \sum_{t=1}^T \pi_{\theta^t}$ is different from $\pi_{1/T \sum_{t=1}^T \theta^t}$ since the model is highly non-linear. Currently, the only method we know to sample from the former is by maintaining all the LLMs simultaneously and selecting one at random to generate a response.
>
> ---
>
> > “Why does the alignment objective in (3) make sense because there is no KL regularized term in the objective? The authors have mentioned that the objective without KL can no longer achieve robust alignment…”
>
> In our manuscript, we stated that the objective with the KL regularization can no longer achieve robust alignment (Lines 152 - 154). Please see the next sections of our response for our answer to this question.
>
> ---
>
> > “How win rate is calculated? Is it via GPT-4?”
>
> We calculate the win rate between two competing models using Llama-3-OffsetBias-8B to compare their outputs. Given the detailed description of this evaluation setup in Line 454, we find it difficult to see why this question is considered a weakness of our submission.

---

> ### Author Response · Authors · 2024-11-21
> **Response to Reviewer SZrT [2/4]**
>
> **Response concerning the necessity of staying close to the SFT policy**
>
> We believe that a substantial part of the criticism in the review centers around the claim that staying close to the SFT policy is crucial for alignment algorithms (“straying closer to the base model via KL divergence is one of the most critical properties of any alignment method. It does not make sense to solve a problem without it.”) and that our proposed training objective lacks such regularizations (“Why does the alignment objective in (3) make sense because there is no KL regularized term in the objective?”). Therefore, we wish to address this criticism before responding to the more detailed questions.
>
> 1. We would appreciate references to evidence or published work that supports the claim that “straying closer to the based model via KL is one of the most important properties of any alignment method.” In fact, recent studies show that KL regularization does not alter the reward frontier of the trained policy [1]. Furthermore, there are effective KL-penalty-free alignment methods, such as SimPO [2], which perform comparably to or better than their counterparts (e.g., DPO). Even when a KL penalty is included in the training objective, its weight tends to be minimal; for instance, InstructGPT [3] sets this weight ($\beta$) at just 0.02.
>
> 2. COMAL guarantees that every iterate is not further away from the Nash equilibrium than the SFT policy. If the SFT policy is already near the Nash equilibrium, our method ensures that the converged policy remains close to the SFT policy. Specifically, we prove that COMAL guarantees $KL(\pi^t || \pi^*)$ is decreasing during the training process (Theorem 1). So if the SFT policy $\pi_{sft}$ is close to the Nash equilibrium $\pi^*$, meaning $KL(\pi_{sft} || \pi^*)$ is small, then it implies that $\pi^t$ also stays close to the SFT policy during training. Importantly, this is achieved without adding a KL penalty to the objective.
>
> 3. On the other hand, if the SFT policy is far from optimal, which is common in many practical alignment fine-tuning settings, we argue that staying too close to the SFT policy can actually be detrimental, imposing an upper bound on performance. Two pieces of evidence support this claim: (a) Our method outperforms SFT not only on the in-domain test set but also on generic benchmarks (results included in the table below). Such improvements will not be possible if the trained policy has to stay close to the SFT policy; (b) Our experiment results show that COMAL beats INPO, which keeps a KL regularization term. This result further validates that adding the KL penalty to the objective is unnecessary for an alignment method to be effective. Our proposed method, COMAL, offers a unique advantage over previous algorithms in this context, as our objective allows the policy to surpass the limitations of the SFT policy and converge to the Nash equilibrium.
>
>
> | Method    | BBH    | GSM8K    | HumanEval   | MMLU   | Avg    |
> |----------|--------|--------|-------------|--------|--------|
> | SFT      | 0.3833 | 0.4850 | 0.5730      | 0.5273 | 0.4921 |
> | COMAL    | 0.4037 | 0.5500 | 0.6100      | 0.5244 | 0.5220 |
>
> References
>
> [1] Gao, Leo, John Schulman, and Jacob Hilton. "Scaling laws for reward model overoptimization." International Conference on Machine Learning. PMLR, 2023.
>
> [2] Meng, Yu, Mengzhou Xia, and Danqi Chen. "Simpo: Simple preference optimization with a reference-free reward." NeurIPS 2024.
>
> [3] Ouyang, Long, et al. "Training language models to follow instructions with human feedback." NeurIPS 2022

---

> ### Author Response · Authors · 2024-11-21
> **Response to Reviewer SZrT [3/4]**
>
> **Detailed response [1/2]**
>
> > “For motivation, the authors have argued that transitivity may not hold in practice, while in theory, it makes sense. However, it would be nice to provide some evidence of such behavior in the real-world preference datasets available in practice.”
>
> Thank you for the feedback. As remarked in [1], there is a long line of research showing that human preference exhibits intransitivity (see, e.g., [2, 3, 4]). It is not easy to test intransitivity in large-scale datasets. However, experiments [4] were conducted with humans, showing that human preference is intransitive. Moreover, even if each human individual has a transitive preference, the resulting expected preference model could still exhibit intransitive preferences (Example provided in [1, Appendix C.2]).
>
> Moreover, intransitiveness is not the only limitation of BT models. BT models are strictly less power than general preference models in representing human preferences. Evidence shows that training with a preference model leads to better performance than BT models (e.g.,  [5]).
>
> To test intransitivity, we would require datasets where users input the same prompts, accompanied by a predefined set of three responses. These responses would then need to be evaluated in a pairwise manner to assess their intransitive relationships—a condition not met in the datasets currently available. However, as real-world preference datasets are collected from human preference, we expect them to exhibit such intransitivity behavior.
>
> References
>
> [1] Munos, Remi, Michal Valko, Daniele Calandriello, Mohammad Gheshlaghi Azar, Mark Rowland, Zhaohan Daniel Guo, Yunhao Tang et al. "Nash Learning from Human Feedback." In Forty-first International Conference on Machine Learning.
>
> [2] Tversky, Amos. "Intransitivity of preferences." Psychological review 76.1 (1969): 31.
>
> [3] Klimenko, Alexander Y. "Intransitivity in theory and in the real world." Entropy 17.6 (2015): 4364-4412.
>
> [4] May, Kenneth O. "Intransitivity, utility, and the aggregation of preference patterns." Econometrica: Journal of the Econometric Society (1954): 1-13.
>
> [5] Ye, Chenlu, Wei Xiong, Yuheng Zhang, Nan Jiang, and Tong Zhang. "A theoretical analysis of nash learning from human feedback under general kl-regularized preference." arXiv preprint arXiv:2402.07314 (2024).
>
> ---
>
> > “Further, to deal with this issue, a general preference optimization framework is utilized, as proposed in existing literature, but this has some issues. While collecting the preferences offline from humans makes sense, it is extremely hard to realize the assumption of having access to preference oracle. Are there any real-world examples of such oracles? For the experiments in this work, an LLM is used for preferences, which are AI-generated only then. Does it mean the framework of general preferences is only realistic with AI feedback not human?”
>
> Thank you for the question.
>
> 1. We note that the preference oracle we used is trained to mimic human preferences using datasets containing human annotations. Therefore, our approach adheres to the standard RLHF setting, where a Bradley-Terry reward model is first trained using human preferences and then used to supervise the training of the policy model. This is also a common approach adopted in recent work to bypass the difficulty of collecting new human annotations [1][2].
>
> 2. Therefore, we believe using an automatic preference oracle is primarily a matter of resources and logistics rather than a limitation of the framework itself. With sufficient resources, involving human participants in pairwise comparisons for preference optimization is feasible. For example, the post-training of Llama 3 introduces multi-round DPO training, where new preference annotations are collected in each round [3].
>
> References
>
> [1] Meng, Yu, Mengzhou Xia, and Danqi Chen. "Simpo: Simple preference optimization with a reference-free reward." NeurIPS 2024.
>
> [2] Dong, Hanze, et al. "Rlhf workflow: From reward modeling to online rlhf." TMLR 2024.
>
> [3] Dubey, Abhimanyu, et al. "The llama 3 herd of models." arXiv preprint arXiv:2407.21783 (2024).

---

> ### Author Response · Authors · 2024-11-21
> **Response to Reviewer SZrT [4/4]**
>
> **Detailed response [2/2]**
>
> > “The contributions are incremental because the authors have just utilized the prox operator, which is well-defined and developed in the optimization theory. Are there any additional challenges in applying that to the alignment context?”
>
> We’d like to clarify that our main contribution is introducing the conceptual prox algorithm to LLM alignment and developing the first last-iterate convergent algorithm, COMAL. Several recent algorithms, including RLHF, DPO, IPO, SPPO, and INPO, use the prox operator for LLM alignment, although not explicitly stated sometimes. Unfortunately, the main challenge has been that these algorithms *either diverge or converge only in a modified game*. To sum up, using the prox operator does __not__ guarantee last-iterate convergence.
>
> COMAL uses the prox-operator in a more sophisticated way inspired by the conceptual-prox algorithm and guarantees last-iterate convergence. We have also conducted experiments to corroborate the theory and validate the effectiveness of COMAL. Hence, we believe our paper introduces a new algorithm in the context of LLM alignment and makes a solid contribution that bridges optimization theory, game theory, and LLM alignment.
>
> ---
>
> > “Why does the alignment objective in (3) make sense because there is no KL regularized term in the objective? The authors have mentioned that the objective without KL can no longer achieve robust alignment. We note that straying closer to the based model via KL is one of the most important properties of any alignment method. It does not makes sense to solve a problem without it.”
>
> Please see the response concerning the necessity of staying close to the SFT policy above.
>
> ---
>
> >“The proposed algorithm again solves a regularized problem only in the algorithm; it is confusing and hard to follow what is the exact approach. I see the authors have proposed to change the reference model to new pi^{t+1}, but this is weird because the final optimal model can be too far from the based model SFT, which is not required at all. Note that the alignment problem is just not only to maximize the preference provided by the preference oracle, but it is also important to stay closer to the base SFT model.”
>
> Here, we clarify our algorithm again. As detailed in Algorithm 1 of our paper, each iteration involves the following two steps: (1) finding the regularized Nash equilibrium of the KL-regularized game with respect to the reference policy using Mirror Descent (MD), as outlined in Algorithm 2, and (2) updating the reference policy to the Nash equilibrium of this regularized game. By iteratively updating the reference policy, our approach establishes equivalence to the conceptual prox algorithm, as per [Nemirovski 2004], ultimately ensuring convergence to the exact Nash equilibrium (Theorem 1).
> Please refer to the response concerning the necessity of staying close to the SFT policy above.
>
> Reference:
>
> [Nemirovski, 2004] Nemirovski, Arkadi. "Prox-method with rate of convergence O (1/t) for variational inequalities with Lipschitz continuous monotone operators and smooth convex-concave saddle point problems." SIAM Journal on Optimization 15.1 (2004): 229-251.
>
> ---
>
> > “Also, the win rate plot without showing KL divergence to the SFT model is an incomplete description of the alignment performance of the approach. See the following alignment papers for that:...”
>
> We thank the reviewer for the comment.
>
> 1. A plot of the KL divergence versus the win rate would be cost-prohibitive, since all the methods we compared are iterative and producing such a plot would require multiple runs of the iterative optimization with different hyperparameters. This cost factor differs significantly from the work recommended on decoding methods.
> 2. We believe such a plot is not necessary to evaluate the alignment approach. To begin with, many published works in this area do not include such plots [1][2][3], including the InstructGPT paper [4]. Moreover, we find that our method outperforms SFT not only on the in-domain test set but also on generic benchmarks (results shown in the table below). We believe these benchmarks are more direct measurements of model performance than the KL divergence to the SFT model.
>
> | Method    | BBH    | GSM8K    | HumanEval   | MMLU   | Avg    |
> |----------|--------|--------|-------------|--------|--------|
> | SFT      | 0.3833 | 0.4850 | 0.5730      | 0.5273 | 0.4921 |
> | COMAL    | 0.4037 | 0.5500 | 0.6100      | 0.5244 | 0.5220 |
>
>
> References
>
> [1] Meng, Yu, Mengzhou Xia, and Danqi Chen. "Simpo: Simple preference optimization with a reference-free reward." NeurIPS 2024.
>
> [2] Dong, Hanze, et al. "Rlhf workflow: From reward modeling to online rlhf." TMLR 2024.
>
> [3] Ivison, Hamish, et al. "Unpacking DPO and PPO: Disentangling Best Practices for Learning from Preference Feedback." NeurIPS 2024.
>
> [4] Ouyang, Long, et al. "Training language models to follow instructions with human feedback." NeurIPS 2022

---

### Official Review · Reviewer_VZSX · 2024-11-06

**Soundness:** 3
**Presentation:** 3
**Contribution:** 3
**Rating:** 6
**Confidence:** 3

**Summary:**

The authors propose to generalize the intrinsic assumptions behind the Bradley-Terry preference model to make the alignment objective more robust to general preference. They propose to formulate the robust alignment problem as a Nash equilibrium solving problem for a two-player zero-sum game. Strong last-iterate convergence results are established. In practice, authors suggest that such an objective can be combined with many existing alignment algorithms.

**Strengths:**

The robust alignment objective, when reduced to prox operator, builds connection with various existing alignment algorithms. The benefit of modifying those algorithms to the proposed COMAL is that it enjoys theoretical last-iterative guarantee while the original algorithms do not. On common alignment benchmark, COMAL shows promising results.

**Weaknesses:**

The weakness mainly lies in the experiment section. According to the formulation of COMAL objective, an equilibrium is found mean the resulting policy wins against any other policy. The presented experiment results testify this. However, my concern is that such an objective may result in heavier alignment tax, which is commonly observed from other alignment algorithm literature.

- To make the meta-algorithm more appealing to real applications, I suggest the authors to also include the evaluation results on academic benchmarks such as logical reasoning, maths, and coding.

**Questions:**

- How is the regularization parameter searched?

- How sensitive is the result to any of the hyperparameters? Did you find any recommended hyperparameter defaults that work well across different base models and training datasets?

---

> ### Author Response · Authors · 2024-11-21
> **Response to Reviewer VZSX**
>
> Thank you for your helpful comments and for recognizing the strengths of our work.
>
> > “The weakness mainly lies in the experiment section. According to the formulation of COMAL objective, an equilibrium is found mean the resulting policy wins against any other policy. The presented experiment results testify this. However, my concern is that such an objective may result in heavier alignment tax, which is commonly observed from other alignment algorithm literature. To make the meta-algorithm more appealing to real applications, I suggest the authors to also include the evaluation results on academic benchmarks such as logical reasoning, maths, and coding.”
>
> We appreciate your suggestion. We have conducted evaluations accordingly using four benchmarks, BigBench Hard (BBH) for reasoning, GSM8K for math problem solving, HumanEval for programming, and MMLU for multitask language understanding:
>
> | Method    | BBH    | GSM8K    | HumanEval   | MMLU   | Avg    |
> |----------|--------|--------|-------------|--------|--------|
> | SFT      | 0.3833 | 0.4850 | 0.5730      | 0.5273 | 0.4921 |
> | Iter-IPO | 0.4065 | 0.5350 | 0.6187      | 0.5249 | 0.5213 |
> | INPO-R3  | 0.4148 | 0.5250 | 0.6130      | 0.5231 | 0.5190 |
> | COMAL    | 0.4037 | 0.5500 | 0.6100      | 0.5244 | 0.5220 |
>
>
> The results show that our algorithm’s (COMAL) performance is comparable to other baselines and does not introduce significant alignment tax. It achieves a particularly strong performance on GSM8K and also the best average performance. We will include this result in the revised version.
>
> ---
>
> > “How is the regularization parameter searched?”
>
> We have discussed the regularization parameter searching process in Section 5.1, Paragraph “Hyper-Parameters” (Lines 469 - 476) and Appendix H:
>
> 1. For the regularization parameter in DPO and IPO ($\eta^{-1}$), we conducted a grid search within the range of 0.001 - 0.02, as detailed in Table 3 in Appendix H. We observed that IPO's performance reaches its optimum and remains relatively stable between 0.002 and 0.01.
>
> 2. For the extra regularization parameter $\tau$ used in INPO and COMAL, we follow INPO's suggestion by setting $\eta\tau$ to a fixed ratio of 1/3 (Lines 472 - 474). Due to the high computational costs associated with the iterative nature of these algorithms (as we noted in Lines 461-465, it takes around 700 GPU hours to finish one training process of an iterative optimization algorithm), we did not conduct a fine-grained search for this ratio. Our preliminary experiments indicated that the algorithm's performance stayed relatively stable within the range of 0.1 - 0.5. We will add more details in the revised version.
>
> ---
>
> > “How sensitive is the result to any of the hyperparameters? Did you find any recommended hyperparameter defaults that work well across different base models and training datasets?”
>
> Thank you for your question! Regarding the sensitivity to hyperparameters, please refer to our response to your earlier question – the recommended value for $\eta^{-1}$ is within the range of 0.002 - 0.01, and the value of $\tau$ can be determined using the ratio of $\eta\tau$ set to 1/3. The variance of performance is relatively low when $\eta^{-1}$ is set to the range of 0.002 - 0.01, and $\eta\tau$ is set to the range of 0.1 - 0.5. Due to the high computational cost, we only experimented with one base model on one dataset. We believe that our observations are likely to generalize well to other datasets, given that the dataset we used includes a wide range of instruction types. We plan to extend our experiments to another model to enhance the generalizability of our results. We will include additional discussion in the revised version of our paper.

---

### Author Response · Authors · 2024-11-27
**Message to Reviewers and the Area Chair: Summary of Revisions**

We thank the reviewers again for their comments and suggestions. We have updated our submission to incorporate these and reflect our response. Below, we summarize the revisions made (all new changes in the submission are highlighted in maroon).

1. **Revision to Appendix A “Related Work”**: Based on our discussion with Reviewer tugW, we provide additional discussion of related work [1,2], which was originally cited and discussed in the main text. We also provide further discussion regarding the contribution of our work in the context of related work: we believe we are the first to introduce the well-established conceptual prox method [3] from the optimization community to alignment fine-tuning,  offering new insights into the alignment problem.

2. **Revision to Appendix H regarding hyperparameter search**: Based on the comment of Reviewer VZSX, we provide more details regarding the preference optimization algorithms’ sensitivity to the hyperparameters.

3. **Added section: Appendix I “Additional Evaluation Results”**: Based on the suggestion of Reviewer VZSX and our discussion with Reviewer tugW, we provide additional evaluation results for the checkpoints trained using COMAL and baseline algorithms in Section 5 “Real-World Experiments”.
   - Appendix I.1 contains the models’ performance on 4 standard LLM benchmarks.
   - Appendix I.2 contains the models’ performance on AlpacaEval using GPT-4 as the preference oracle/evaluator.
The added results further demonstrate the effectiveness of COMAL.

4. **Added section: Appendix J “More Practical Implementations of COMAL”**: Based on our discussion with Reviewer tugW, we provide more practical implementations of COMAL using the SPPO loss, the DRO loss, and the REBEL loss, using their extended forms as shown in Appendix E.

We would also like to kindly remind the reviewers to review our responses and the revised submission. We would appreciate your further feedback and a re-evaluation of our submission.

References

[1] Perolat, Julien, et al. "From poincaré recurrence to convergence in imperfect information games: Finding equilibrium via regularization." International Conference on Machine Learning. PMLR, 2021.

[2] Abe, Kenshi, et al. "Adaptively Perturbed Mirror Descent for Learning in Games." Forty-first International Conference on Machine Learning. 2024

[3] Nemirovski, Arkadi. "Prox-method with rate of convergence O (1/t) for variational inequalities with Lipschitz continuous monotone operators and smooth convex-concave saddle point problems." SIAM Journal on Optimization 15.1 (2004): 229-251.

---

### Meta-Review · Area_Chair_GYu5 · 2024-12-23

**Metareview:**

Most existing general preference-alignment methods only achieve average-iterate convergence to the Nash equilibrium policy, rather than last-iterate convergence. Inspired by convergent algorithms in game theory, the authors propose a novel method, COMAL, which formulates the robust alignment problem as a Nash equilibrium problem and guarantees last-iterate convergence.

This paper proposes an algorithm that can provide last iterate guarantee for many existing alignment algorithms. However, as challenged by the reviewers, there are some weaknesses in the experimental parts and the algorithm’s novelty. I would suggest that the authors write more clearly to help readers fully understand the main contribution of this paper.

**Additional Comments On Reviewer Discussion:**

* In response to Reviewer tugW’s request for additional related work, the authors revised Appendix A.
* To address Reviewer VZSX’s concern about the sensitivity of preference optimization algorithms to hyperparameters, the authors revised Appendix H.
* Both Reviewer tugW and Reviewer VZSX requested additional evaluation results for the checkpoints trained with COMAL and baseline algorithms; these updates can be found in Appendix I.
* Reviewer tugW requested more real world application, the authors add Appendix J for more practical implementation discussion.

---

### Decision · Program_Chairs · 2025-01-22

Reject